# Critical Assessment of MetaProteome Investigation (CAMPI): a multi-laboratory comparison of established workflows

Tim Van Den Bossche [1,2,23], Benoit J. Kunath [3,23], Kay Schallert [4,23], Stephanie S. Schäpe[5,23], Paul E. Abraham[6], Jean Armengaud [7], Magnus Ø. Arntzen [8], Ariane Bassignani [9], Dirk Benndorf [4,10,11], Stephan Fuchs [12], Richard J. Giannone[6], Timothy J. Griffin [13], Live H. Hagen [8], Rashi Halder [3], Céline Henry[9], Robert L. Hettich [6], Robert Heyer [4], Pratik Jagtap [13], Nico Jehmlich [5], Marlene Jensen [14], Catherine Juste[9], Manuel Kleiner [14], Olivier Langella [15], Theresa Lehmann[4], Emma Leith[13], Patrick May [3], Bart Mesuere [1,16], Guylaine Miotello[7], Samantha L. Peters [6], Olivier Pible [7], Pedro T. Queiros[3], Udo Reichl[4,11], Bernhard Y. Renard [12,17], Henning Schiebenhoefer [12,17], Alexander Sczyrba [18], Alessandro Tanca [19], Kathrin Trappe[12], Jean-Pierre Trezzi[3,20], Sergio Uzzau[19], Pieter Verschaffelt [1,16], Martin von Bergen [5], Paul Wilmes [3,21], Maximilian Wolf[4], Lennart Martens [1,2,24✉] & Thilo Muth [22,24]

Metaproteomics has matured into a powerful tool to assess functional interactions in microbial communities. While many metaproteomic workflows are available, the impact of method choice on results remains unclear. Here, we carry out a community-driven, multi-laboratory comparison in metaproteomics: the critical assessment of metaproteome investigation study (CAMPI). Based on well-established workflows, we evaluate the effect of sample preparation, mass spectrometry, and bioinformatic analysis using two samples: a simplified, laboratory-assembled human intestinal model and a human fecal sample. We observe that variability at the peptide level is predominantly due to sample processing workflows, with a smaller contribution of bioinformatic pipelines. These peptide-level differences largely disappear at the protein group level. While differences are observed for predicted community composition, similar functional profiles are obtained across workflows. CAMPI demonstrates the robustness of present-day metaproteomics research, serves as a template for multi-laboratory studies in metaproteomics, and provides publicly available data sets for benchmarking future developments.

A full list of author affiliations appears at the end of the paper.

Microbial communities play a primary role in global biogeochemical cycling and form complex interactions that are crucial for the development and maintenance of health in humans, animals, and plants. To fully understand microbial communities and their interplay with their environment requires knowledge not only of the microorganisms involved and their biodiversity, but also of their metabolic functions at both the cellular and community level[1]. As proteins constitute the key operational units performing these functions, metaproteomics has emerged as the most relevant approach to characterize the functional expression of a given microbiome[2,3]. Metaproteomics corresponds to the large-scale characterization of the entire set of proteins accumulated by all community members at a given point in time, known as the metaproteome[4]. Since its first introduction in 2004[5], mass spectrometry (MS)-based metaproteomics has quickly emerged as a powerful tool to functionally characterize a broad variety of microbial communities in situ. This allows a direct link to the phenotypes on a molecular level and shows the adaptations of the microorganisms to their specific environment[6]. Metaproteomics thus complements other meta-omic approaches such as metagenomics and metatranscriptomics, as these only have the exploratory power to assess the diversity and functional potential of microorganisms, but cannot observe their actual phenotypes[7].

In metaproteomics, proteins are commonly measured using the shotgun proteomics approach. Here, the proteins are subsequently extracted, isolated, and digested into peptides, after which these are separated and analyzed using liquid chromatography coupled to tandem mass spectrometry (LC–MS/MS). The obtained MS/MS spectra are then matched against in silico generated spectra derived from a protein sequence database, leading to peptide spectrum matches (PSMs). Hereafter, the identified peptides are used to infer the proteins present in the sample. Proteins can then be annotated with taxa and functions, providing information on gene expression levels[8].

Each of the aforementioned steps can potentially influence the outcomes of a metaproteomic analysis and every step brings specific benefits as well as challenges. As a result, multiple workflows have been established. While such diversity brings flexibility, it also complicates the comparison of results across different experiments. Sample processing challenges include protein recovery due to the presence of different matrices[9], the presence of different types of microorganisms with different optimal lysis conditions[10,11], and limited depth of analysis[3] and quantification[12] due to an increased sample complexity. Environmental samples, such as feces or soil, are complex mixtures that can contain microbial cells, host cells, plant-derived fibrous materials, and other abiotic components. Therefore, the composition and abundance of these components must be considered when choosing an appropriate method for cellular lysis and protein extraction. Fortunately, the most commonly used methods nowadays are relatively robust, and generally provide a reasonably representative extraction of proteins found in these complex mixtures. However, because differences exist, methods still need to be optimized for the specific samples and projects[13,14] Besides, apart from different sample processing protocols, different mass spectrometers might also lead to a variation in results.

Moreover, metaproteomics comes with many specific bioinformatic challenges[8,15]. First, the choice of an appropriate sequence database is critical for peptide identification[16,17]. Typically, large databases can strongly impact sensitivity and false discovery rate (FDR) estimation[18], while incomplete reference databases can lead to missing or false positive identifications[19,20]. Second, the protein inference problem[21] is more pronounced in metaproteomics due to many homologous proteins from closely related organisms[22]. As a result, several dedicated bioinformatic tools have been developed or extended for metaproteomic analysis[23–30]. Despite these challenges, the added value of metaproteomics has already been demonstrated in numerous examples from both the environmental and medical fields, providing unprecedented insights into the functional activity of microbial communities[7,22,31–43].

Nevertheless, a lingering concern is the potential risk of unintended, approach-based biases inherent in various metaproteomic workflows. This is important because reproducibility is key to translate metaproteome studies into applications (e.g., clinical or industrial). Consequently, a comprehensive evaluation of widely used workflows is required to assess their respective outcomes. In the past, various reference data sets from defined microbial community samples (i.e., for which the comparison of established workflows composition is known a priori) have been used in individual benchmarking studies[44–46]. However, a ring trial with different laboratories involved has not yet been performed in the field of metaproteomics.

To fill this gap, the 3rd International Metaproteomics Symposium (December 2018, Leipzig, Germany) hosted a multi-laboratory benchmarking study in the form of a community challenge. Participating laboratories received two microbial samples: a simplified mock community simulating the gut microbiome (SIHUMIx) and a complex, natural stool sample (fecal sample). Each group was allowed to use any preferred sample preparation, analysis, and data evaluation pipeline.

Here, we describe the results of this community-driven study, referred to as the Critical Assessment of MetaProteome Investigation (CAMPI). We compare and discuss the employed workflows covering all analysis steps from sample preparation to the bioinformatic identification and quantification. Moreover, we compare the metaproteome results with sequencing read-based analyses (metagenomics and metatranscriptomics). We found that meta-omics databases performed better than public reference databases across both samples. More importantly, even though larger differences were observed in identified spectra and unique peptide sequences, the different protein grouping strategies and the functional annotations provided similar results across the provided data sets from all laboratories. When minor differences could be observed, these were largely due to differences in sample processing methods and partially to bioinformatic pipelines. Finally, for the taxonomic comparison, we found that overall profiles were similar between read-based methods and proteomics methods, with few exceptions. Apart from these immediate conclusions, the CAMPI study also delivers highly valuable benchmark data sets that can serve as a foundation for future method development for metaproteomics.

## Results

At the 3rd International Metaproteome Symposium in December 2018, individual laboratory outcomes of a collaborative, multi-laboratory effort to compare metaproteomic workflows were presented. In this study, metaproteomics data was acquired in seven laboratories, using a variety of well-established platforms. Figure 1 provides a general overview of the study design showing (i) the provision of two types of samples (SIHUMIx and fecal) to the study participants, (ii) the various experimental workflows of biomolecule extraction and MS/MS acquisition, and (iii) the bioinformatic processing steps from protein database generation to database search identification and follow-up analyses (more details in the "Methods" section, see Supplementary Data 1 for an overview of all methods).

At the Symposium, the decision was made to re-analyze the acquired data with different bioinformatics pipelines, to obtain a

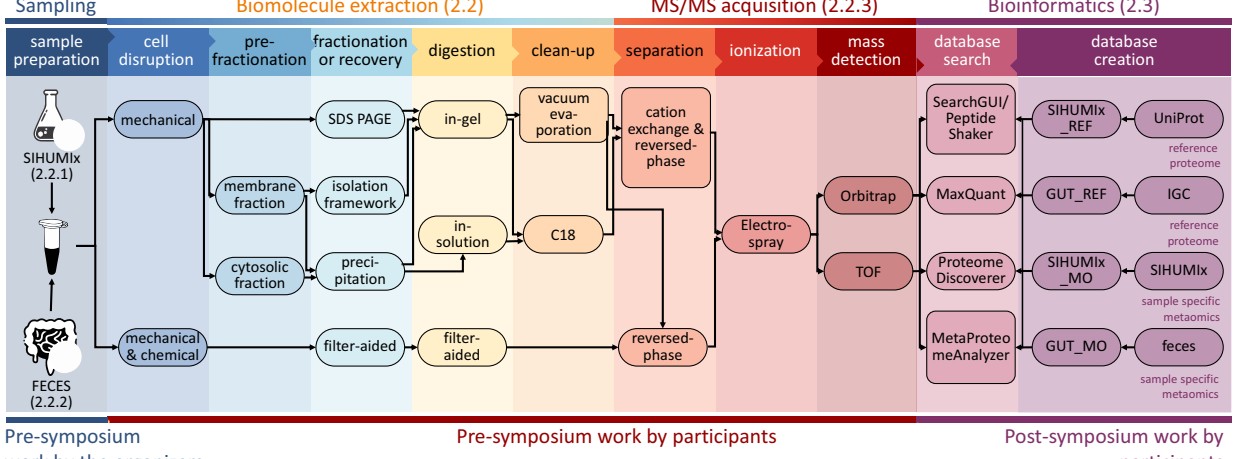

**Fig. 1 Schematic representation of the main sample preparation steps and follow-up analyses of the CAMPI study.** The figure consists of three parts: (i) Pre-symposium work by the organizers (left panel). The two samples (SIHUMIx and fecal sample) were, prior to the symposium, aliquoted and distributed over the participating laboratories. (ii) Pre-symposium work by participants (middle panels). Every used method by the participants, going from cell disruption to mass detection, is displayed. (iii) Post-symposium work by participants (right panel). The bioinformatics analyses, i.e., database creation and database search for peptide and protein identification, were harmonized to make the results between all participating laboratories comparable. The stock icons in the leftmost column were obtained from vecteezy.com, flaticon.com, and labicons.net.

multi-laboratory effort in metaproteomics to independently evaluate available methodological and computational approaches, in line with similar community-driven benchmarking studies[47–50]. In the first "Results" section, we analyzed 42 raw files (21 for the SIHUMIx sample and 21 for the fecal sample) from 24 different workflow combinations with X!Tandem using either public or in-house generated protein databases (see Fig. 1 for a general overview, and Fig. 2 for the results; see online Methods section for the database construction). A more in-depth comparison of sample preparations, bioinformatic pipelines, and taxonomic and functional annotations using a sub-selection of ten data sets is available after the first "Results" section.

**Complex sample processing workflows and sample-specific meta-omic search databases lead to more identifications**. In order to study the effect of the different sample processing and LC–MS/MS workflows on the identification outcome, we searched all submitted MS files using the widely used X!Tandem search engine[51]. To investigate the influence of the chosen database, we searched each file against a publicly available reference database (SIHUMIx_REF and GUT_REF) and against a multi-omic database (SIHUMIx_MO and GUT_MO). The comparison of all CAMPI workflows is displayed in Fig. 2 (raw data in Supplementary Data 2).

The results greatly differed between the samples and workflows in terms of absolute numbers of acquired spectra, identified spectra, and relative amount of identified spectra (identification rates). For the SIHUMIx data set, the number of acquired spectra varied between 47k to 260k, and identification rates varied between 29.99% and 68.64% for SIHUMIx_REF and between 32.52% and 73.34% for SIHUMIx_MO. For the fecal data set, between 44k and 223k spectra were acquired, with identification rates between 11.99% and 34.79% for GUT_REF, and between 15.70% and 41.94% for GUT_MO.

The differences in acquired spectra show a clear relation to the method used, as similar methods or replicates show highly similar numbers of acquired spectra. As expected, more complex methods with longer gradient lengths (S03 and S04: 260 min, S05 and S06: 460 min, S08: 240 min, F01: 210 min, F02: 160 min),

fractionation (S11, F07: 4 fractions), and additional separation methods such as MudPIT[52] (F01: 4 fractions) or ion mobility (PASEF)[53] (S13, F09) led to up to eight times more identified spectra, but at the cost of increased time and resources spent[54] (see Supplementary Data 1 for a detailed description, and Supplementary Data 2 for an overview of the samples). Notably, identification rates were not necessarily correlated with the total number of identifications. For example, between analyses S03 and S05, which used a 260 and 460 min LC gradient length, respectively, a higher absolute number of identified spectra was found for the 460 min gradient, but also a lower identification rate. As expected, if an MS instrument is provided with the ability to acquire more spectra, it will do so. However, the gains in spectral acquisition do not readily translate into gains in identification. There is thus a potential for diminishing returns when going for more complex methods. There is also a somewhat consistent drop in the number of acquired spectra of around 10% when comparing SIHUMIx samples with fecal samples for similar workflows (e.g., S09-S10 with F05-F06, and S13 Reps 1-3 with F09 Reps 1-3). However, occasionally this drop is much greater, as for S11_Fract1-4 and F07_Fract1-4. The overall limited drop might be attributed to the higher complexity of the fecal sample, and corresponding ion suppression effects. The differences in identification rate are likely to be derived from the choice of the search database. The identification rates for the publicly available databases were invariably lower, which is due to their larger and less specific search space, consistent with literature[16,18,20,44,55]. Here, these public reference databases (SIHUMIx_REF and GUT_REF) contained 1.6 and 16 times, respectively, more unique in silico digested peptides than the corresponding multi-omic databases (SIHUMIx_MO and GUT_MO) (Supplementary Data 3).

Overall, our results indicate that generating a sample-specific meta-omic database can be advantageous for complex metaproteomics samples, such as the human gut microbiome, and even more so for complex and poorly characterized samples such as soil microbiota. The smaller meta-omic databases require less computational resources (e.g., CPU and RAM) and tend to be more accurate due to their tailored composition. However, for their generation, meta-omic databases require additional experimental and computational resources, and are often not as well

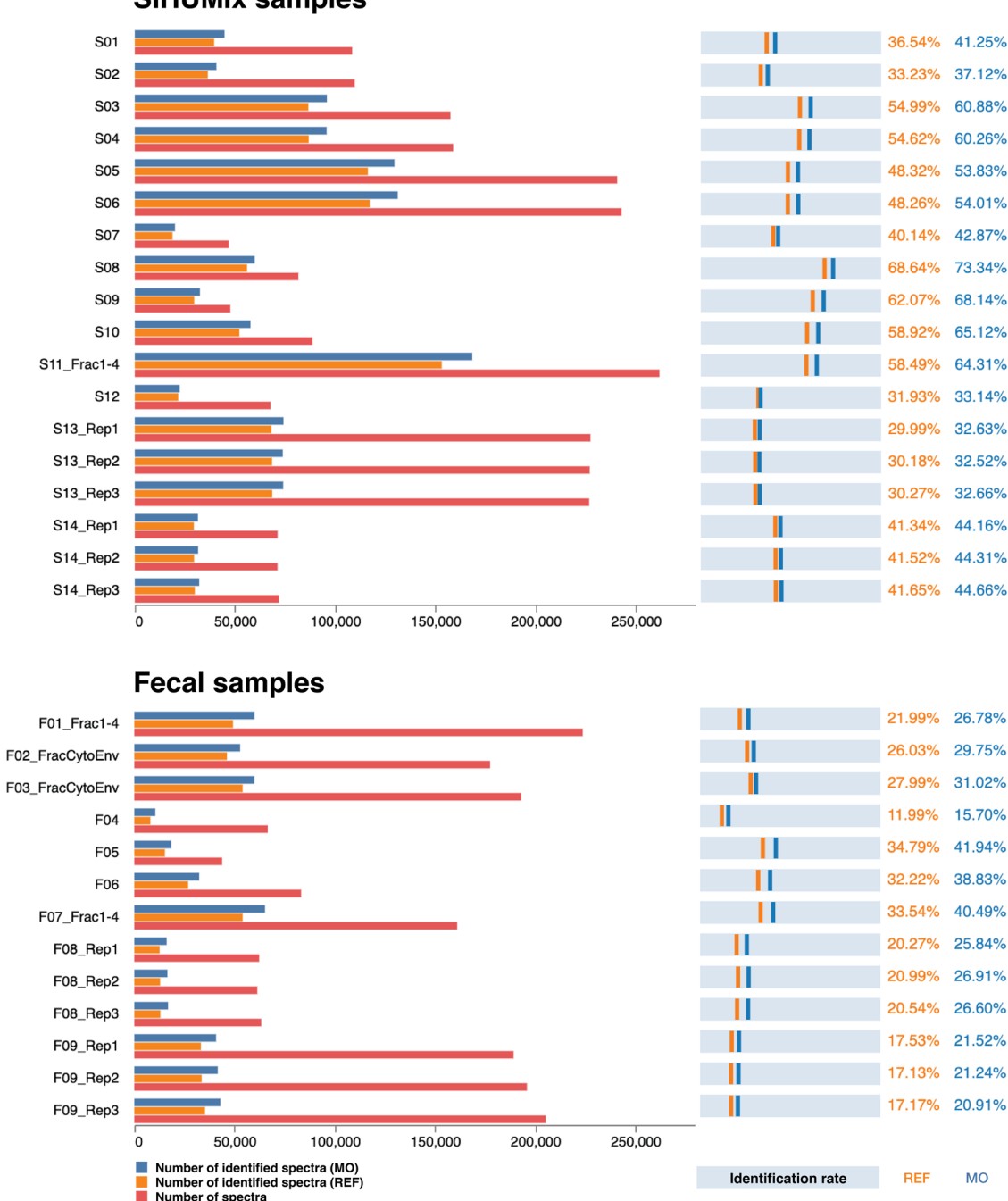

**Fig. 2 Comparison of identification rates across all CAMPI workflows.** On the left side, the bar charts show the number of identified spectra using the reference (REF) database (orange), the number of identified spectra using the multi-omic (MO) database (dark blue) and total amount of measured spectra (red). On the right side, the light blue bars represent the identification rate calculated as the percentage of spectra that yielded a peptide identification at 1% FDR for both the REF database (orange) and the MO database (dark blue). The specific protocols can be found in Supplementary Data 1. For database searching, X!Tandem was used as a single search engine. Source data is provided in Supplementary Data 2.

assembled and/or annotated as reference databases. Because the composition of SIHUMIx was known, the benefit of using a tailored meta-omic database was limited and the analysis was feasible with available reference proteomes. In contrast, the community for the fecal sample was unknown, which represents the typical scenario in metaproteomics.

For known reference samples (such as SIHUMIx), it is, therefore, reasonable to simply use the reference database, while the largely unknown fecal sample community is best analyzed using a tailored meta-omic database. In the following sections, we

thus opted to use only the SIHUMIx_REF and GUT_MO search databases for SIHUMIx and fecal data sets, respectively.

**Different bioinformatic pipelines resulted in highly similar peptide identifications.** To investigate the effect of the bioinformatic pipelines on peptide identification, we compared the two data sets with the most identified peptides (S11 and F07) (Fig. 3). To ensure a robust and reliable comparison, we fixed the search parameters for the four different bioinformatic pipelines employed (see online Methods for details).

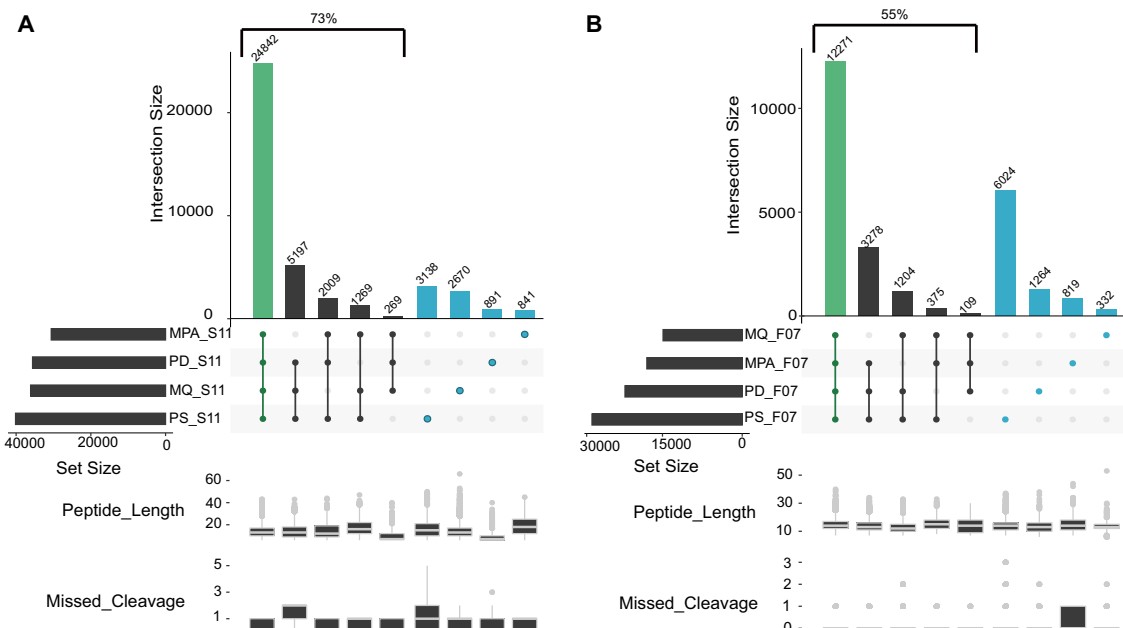

**Fig. 3 UpSet plot comparison of identified sets of peptides using different bioinformatic pipelines.** The left panel displays the results for the SIHUMIx sample S11 (**A**), while the right panel corresponds to the results for the fecal sample F07 (**B**). The four different bioinformatic pipelines (MetaProteomeAnalyzer (MPA, using X!Tandem and OMSSA), Proteome Discoverer (PD, using SequestHT), MaxQuant (MQ, using Andromeda), SearchGUI/PeptideShaker (PS, using X!Tandem, OMSSA, MS-GF+, and Comet)) are indicated on the x-axis and sorted by increasing set size. Set size corresponds to the total number of peptides identified per tool, and intersection size corresponds to the number of shared peptides identified in different approaches. Green highlights the intersection, and blue shows unique peptides to each tool. The lower panel box plots show peptide lengths, and number of missed cleavages for each intersection. Source data is provided as a Source Data file.

For SIHUMIx, the majority of the identified peptides (54.2%) were found by all four bioinformatic pipelines (Fig. 3A), while this ratio dropped to 40% for the more complex fecal F07 sample (Fig. 3B). As expected, this percentage increased to 73% and 55%, respectively, when considering the peptides identified by at least three out of four tools. Interestingly, 16% of the peptides were uniquely identified by a single bioinformatic pipeline for the S11 data set (3138, 2670, 891, and 841 peptides for SearchGUI/PeptideShaker, MaxQuant, Proteome Discoverer, and MPA, respectively), while this was 27% for the F07 data set (6024, 1264, 819, and 332 peptides for the SearchGUI/PeptideShaker, Proteome Discoverer, MPA and MaxQuant pipeline, respectively). The number of search engines varies between pipelines, with one for MaxQuant (Andromeda) and ProteomeDiscoverer (SequestHT), two for MPA (X!Tandem, OMSSA), and four for SearchGUI (X!Tandem, OMSSA, MS-GF+, and Comet). Furthermore, each algorithm uses its own score as a quality metric for finding the best matching peptide for a spectrum. This score varies between the search engines and can even result in different peptide identifications for the same spectrum[56].

Overall, the combination from multiple search engines as performed by SearchGUI/PeptideShaker (four algorithms) resulted in the highest number of identifications, which is in line with the previous studies in proteomics and proteogenomics[57,58]. This effect may be attributable to algorithms with more sophisticated scoring methods (e.g., MS-GF+[59] used in Search-GUI, but not in MPA), which generally lead to more identifications overall. However, we do expect that novel search engines based on machine learning algorithms can still boost the number of peptide identifications in the field of metaproteomics[60].

Additionally, we compared the pipelines in terms of peptide features using the peptide lengths and the number of missed cleavages (lower panels of Fig. 3A, B). While few outliers could be observed (e.g., peptide length over 50 AA for MaxQuant and

missed cleavages over two for SearchGui/PeptideShaker and ProteomeDiscoverer), the features were overall equally distributed between pipelines. Most of the differences thus seemed to be simply linked to the search engines used.

Because the SearchGUI/PeptideShaker combination provided the most identifications, relatively few identifications were missed by excluding the other three pipelines. We therefore preferred to only use the results of the SearchGUI/PeptideShaker pipeline in the following sections, which investigate the effect of different sample processing workflows on downstream peptide identifications. These analyses are performed on ten representative data sets that have been selected based on their type of fractionation and MS instrument. These include six SIHUMIx, and four fecal data sets (Supplementary Data 2).

**Differences between laboratory workflows are mostly attributable to low abundance proteins.** After we ruled out bioinformatic workflows as a source of significant difference between samples, we investigated differences arising from different laboratory workflows. We compared the overlap and uniqueness of identifications at the level of peptides, protein subgroups, and the 50% most abundant protein subgroups for the selected laboratory workflows in Fig. 4. The figure shows how many peptides and protein subgroups are uniquely identified by a single laboratory workflow and how many are identified by all laboratory workflows.

At the peptide level (Fig. 4A, B), more complex workflows, such as those with longer gradient length and fractionation, identified the most peptides in general (as shown earlier in Fig. 2) as well as the most workflow-specific peptides, thus limiting the potential for overlap. The number of identified peptides shared between all workflows was quite limited: only 3557 peptides (4.9% of all identified peptides) in the SIHUMIx data sets, and 2186 peptides (3.4% of all identified peptides) in the fecal data set. At

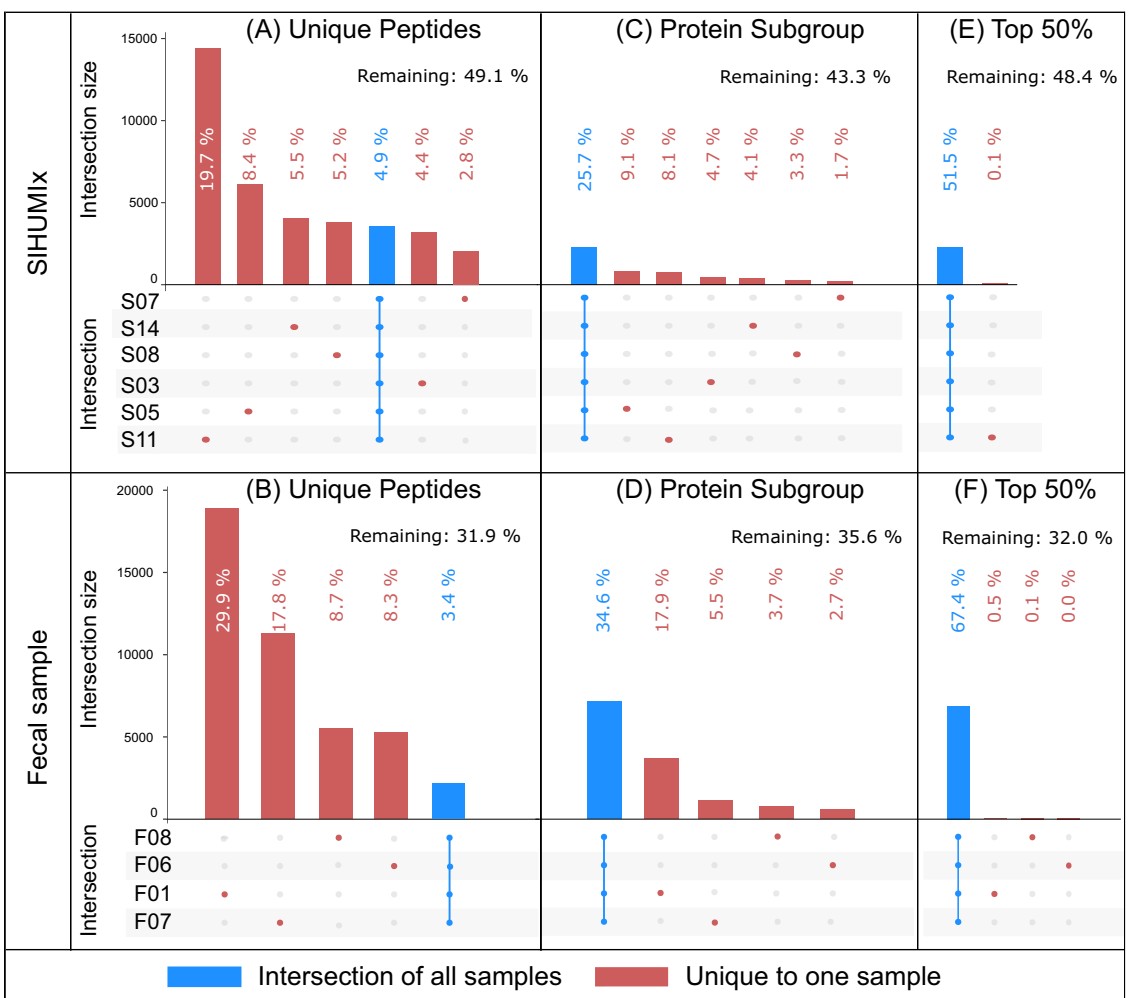

**Fig. 4 UpSet plot comparison of sets of identified peptides, protein subgroups, and 50% most abundant protein subgroups. A**, **B** Identified peptides, **C**, **D** all identified protein subgroups, and **E**, **F** top 50% subgroups (SIHUMIx and fecal sample, respectively). Top 50% protein subgroups were selected in terms of spectral count per subgroup. The figure is based on the identifications obtained using SearchGUI/PeptideShaker. The intersection size displays the number of features shared in an intersection. An intersection corresponds to features shared across multiple samples. This figure only displays features unique to a sample (red dot), and shared across all samples (blue bar overlapping all points). Source data is provided as a Source Data file.

the protein subgroup level (Fig. 4C, D), the intersections of protein subgroups shared across all workflows were 25.7% and 34.6% for the SIHUMIx and fecal data sets, respectively. These percentages increased to 51.5% and 67.4% when we only considered the 50% most abundant protein subgroups (Fig. 4E, F). Large differences between laboratory workflows observed at the peptide level were thus attenuated at the protein subgroup level, and further reduced for the 50% most abundant protein subgroups. This trend was also clearly visible when considering all intersections, including partial agreement among some samples (Supplementary Figs. 2 and 3). Of note is that the data sets that only differed in a single laboratory method parameter, such as LC gradient length (S03 and S05) or fractionation (F06 and F07), showed a much higher overlap. Also, the number of protein subgroups identified uniquely in a single sample mostly disappeared when only considering the 50% most abundant subgroups. We investigated this further by analyzing the agreement between samples at all top-N-% values (Supplementary Fig. 4). A clear trend emerged: the lower the agreement between samples on a given subgroup, the lower the abundance of this subgroup. Furthermore, subgroups that were identified with a single peptide—and therefore usually at the lowest abundance—track very closely with the subgroups identified in only a single

sample. Finally, when considering the actual spectral abundance of subgroups, those subgroups that were found in all samples also explained at least 77% of the identified spectra. It is therefore clear that the low agreement between samples at the peptide level is mostly attributable to the identification of low abundant proteins. The complexity of the samples and the limited speed of mass spectrometers in DDA mode led to stochasticity in precursor selection at the low end of the dynamic range. Low abundant protein subgroups with only one peptide thus behave more like peptides, where stochastic selection causes large differences between samples. It is worth noting that this issue is completely avoided by only selecting the top 50% of protein subgroups. Overall, it can be concluded that while different laboratory workflows provide very different peptide identifications, the protein subgroups are well preserved.

Because protein grouping plays such an important role in translating peptide identifications into biologically meaningful information, we decided to analyze two commonly used grouping methods in more detail. Protein grouping is achieved using the algorithms PAPPSO[61] and MPA[28] (Supplementary Note 3). These two methods use different rules for protein inference: PAPPSO uses Occam's razor, and MPA uses anti-Occam's razor[62]. The first approach provides a minimum set of proteins

that explains the presence of the detected peptides, while the second approach keeps all proteins matched by at least one peptide. Both PAPPSO and MPA can create two types of protein groups: comprehensive groups based on at least one shared peptide, and more specific subgroups based on a complete shared peptide set. Subgroups were deemed more suitable for this analysis, as comprehensive groups collated proteins that were too heterogeneous leading to diverse protein functions within the same group (Supplementary Data 4 and 5). This might not be the case for smaller data sets, as a smaller data set also decreases the chance for peptides that link highly dissimilar proteins together. For the SIHUMIx samples, the two protein grouping methods PAPPSO and MPA provided very similar numbers of both protein groups (8802 and 8769) and subgroups (10,132 and 10,134), while substantial differences were found for the fecal samples (protein groups: 10,063 and 9712; subgroups: 17,576 and 21,973, for PAPPSO and MPA, respectively) (Supplementary Data 6). While cross-sample correlation (Supplementary Figs. 5 and 6) confirmed that the impact of bioinformatic pipelines on the analysis here was negligible, little else could be learned from this correlation analysis. To shed some light on these differences between protein grouping methods, we analyzed the agreement between samples for different grouping approaches (Supplementary Figs. 7 and 8). Notably, when applied to the fecal sample, the protein groups resulted in an unusually high number of groups that are unique to F10. However, it remains unclear which of these approaches is better able to capture the actual composition of the sample, or even if the performance of the approaches varies for different types of samples. Because PAPPSO grouping removes likely wrong identifications from homologs, it could be more appropriate for single-organism proteomics or for taxonomically well-defined samples like SIHUMIx. In contrast, the grouping from MPA could be more appropriate for complex, unknown samples like the fecal sample (where shared peptides become much more likely) as it retains all information for the grouping (Supplementary Note 3). To conclude, both protein grouping methods provide highly similar results for the SIHUMIx sample, but diverge on the fecal sample, likely due to the increased complexity of the protein inference task in the latter.

**Comparison of meta-omic methods reveals differences between peptide and protein-derived analysis of taxonomic community composition.** To determine if differences between sample processing workflows have an effect on the overall biological conclusions, we quantitatively compared the identified taxa for each selected sample from both data sets using spectral counts, and this at the peptide, the protein subgroup, and the sequencing read level.

We found different trends between the SIHUMIx and fecal samples (Figs. 5 and 6). For SIHUMIx, the taxonomic distributions were relatively similar between the metagenomic read, peptide, and protein group levels based on the principal component analysis. Hierarchical clustering highlighted clusters of samples, with the peptide and protein subgroup profiles for samples S07 and S14 clustering with the read-based profile (Fig. 5A and Supplementary Fig 9A, B). Interestingly, samples with more complex sample processing methods (S03, S05, and S08) did not show clustering between the peptide and the protein subgroups level. While species were found to be similar between methods overall, there were some notable differences (Fig. 5B). All methods agreed that *Bacteroides thetaiotaomicron* was the most abundant species, and found *Escherichia coli* at 10–13% abundance. However, differences were found for *Blautia producta*, which was barely found by the proteomics methods, while found at around 5% abundance by metagenomics. It is

interesting to consider that this might be caused by the construction of the reference database: at the moment of construction, the UniprotKB reference proteome of *Blautia producta* was not available, and multiple *Blautia sp.* proteomes were therefore provided instead. When looking at the Unipept results in detail, 15% of the peptides were associated with the genus Blautia (Supplementary Data 7), which indicates that the lower identification of *Blautia producta* at the peptide level is due to difficulties in resolving *Blautia* at the species level, rather than a lack of identified *Blautia* peptides during the metaproteomic search. Additionally, *Clostridium butyricum* was not found by the read-based method, while *Clostridiales bacterium* and *Bacteroides dorei* were falsely found by the protein-centric method as these are not present in the SIHUMIx sample. However, these last two were both found at very low abundance. For completeness, the comparisons of community composition for SIHUMIx at the genus level were added in Supplementary Fig. 10.

For the fecal data set, which was grouped at the family level, relatively distinct assessments of community composition were obtained from the read-based, peptide, and protein subgroup levels (Fig. 6A). While the same families were identified, these had different proportions across methods (Fig. 6B). Metatranscriptomic information (Feces_MT) was available for the fecal sample and RNA and DNA results were closely colocated, while proteins and peptides were spread out from the read-based methods, but also from each other (Fig. 6A). The difference between metagenomics/metatranscriptomics and metaproteomics is not surprising because these different methods highlight community profiles from different angles. As already shown before, metagenomics provides a good assessment of community composition in terms of cell numbers for each species, while metaproteomics reflects proteinaceous biomass for each species[45].

Strikingly, for the fecal samples, the community composition as quantified at the peptide level proved to be more similar to the read-based than to the protein-based composition (Fig. 6A and Supplementary Fig. 11A, B). This discrepancy is likely due to the fundamental issue of protein inference. Indeed, in metaproteomics, identification and quantification usually rely on discriminative peptides. As the data sets get more complex, higher levels of sequence homology for many proteins will be observed and will lead to a much greater level of peptide degeneracy across taxonomies[63]. Direct taxon inference from peptides thus likely results in more stringent taxonomy filtering, due to the necessity to rely only on taxon-specific peptides. In fact, the proportion of unclassified peptides between the SIHUMIx and the fecal samples went up from 24.2 to 73.4% due to the increased taxonomic complexity of the fecal data set. In contrast, the proportion of unclassified protein subgroups went down from 69.9% for SIHUMIx to 9.5% for the fecal samples. This latter difference, while large, is not that surprising because the fecal sample considered protein subgroups at the family level, while the SIHUMIx sample considered protein subgroups at the species level and only considered SIHUMIx species, therefore greatly limiting peptide-level degeneracy. For the fecal sample, proteins within a subgroup are usually associated to the same family, which explains the higher proportion of protein subgroups that can be classified for the fecal samples.

Additionally, regarding quantification, protein grouping for the fecal samples was done using MPA, which includes all peptides (shared as well as unique), while peptide level quantification only took into account taxon-specific peptides. Depending on the sample and the method used, the taxonomic resolution will thus vary. To better illustrate that, we compared the resolution across omes and across protein grouping methods (Supplementary Fig. 12A, B). We see that there is usually a drop of resolution either at the species (SIHUMIx) or the genus (Fecal) level and

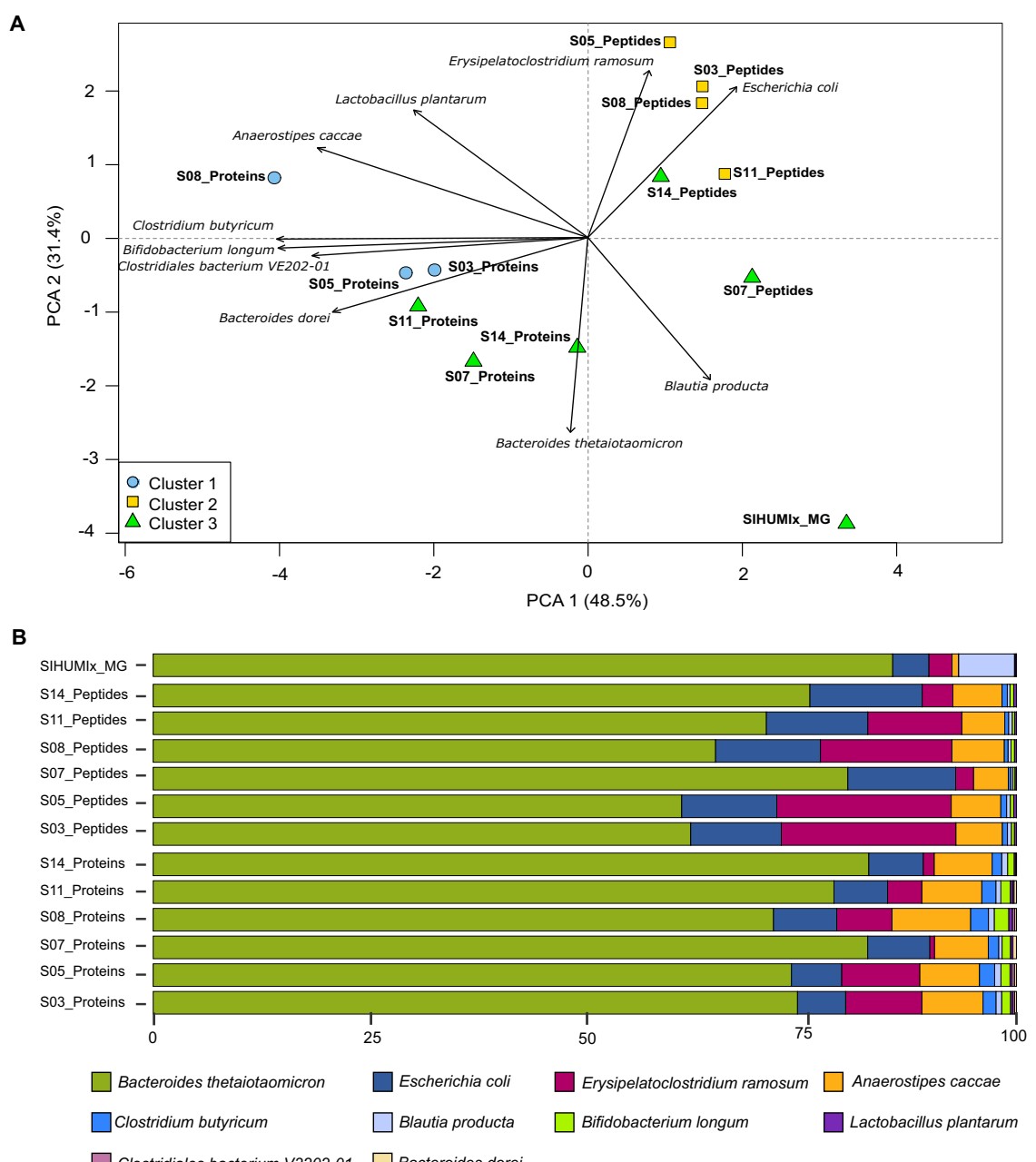

**Fig. 5 Comparisons of community composition for SIHUMIx at the species level.** The upper panel shows PCA clustering of the results (**A**). Different approaches and tools used for taxonomic annotation (MG - mOTU2, Peptides - Unipept, and Proteins - Prophane) are indicated in the label. Clusters ($k = 3$) were calculated using manhattan distance and are represented by blue, yellow, and green. Features not annotated at species level were considered unclassified and discarded for PCA calculation. Unclassified features accounted for 24.2% and 69.9% of data for peptide and protein subgroup levels. Variables driving differences between samples are represented by black arrows. The lower panel details taxonomic profiles of each sample as bar plots (**B**). Source data is provided as a Source Data file.

that the PAPPSO grouping method has a higher resolution for complex samples as already discussed in Supplementary Note 3.

Altogether, the degree of degeneracy at the peptide level combined with the grouping method employed for the proteins leads to a different amount of features used for each analysis and thus to different composition profiles between peptide-centric and protein-centric approaches.

Ultimately, due to the sequence homology issue, worse taxonomic resolution will be available for larger, more complex data sets as illustrated in the differences between the SIHUMIx and the fecal data sets. A promising approach to tackle these limitations can take advantage of shared rather than taxon-

specific peptides (and thus avoiding the previously mentioned issues) to assess the biomass content of a given community[63]. However, regardless of the chosen approach, it is clear that a higher level of peptide coverage will be quite helpful for higher resolution taxonomic annotation, and that metaproteomics will therefore benefit from focusing on analysis depth at the peptide level.

**The functional profile is similar between different metaproteomics workflows.** A major strength of metaproteomics is the ability to provide functional information that reflects the

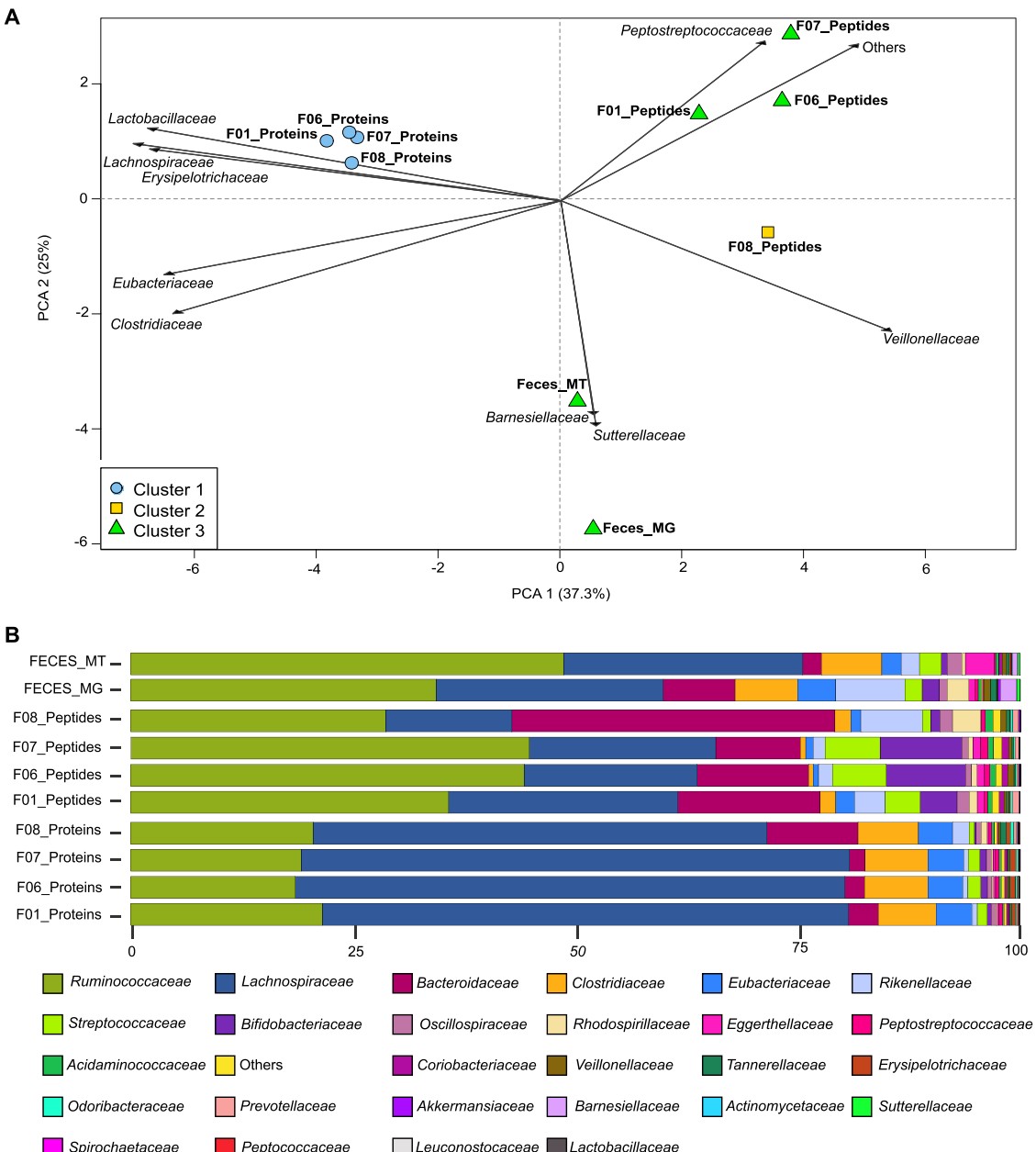

**Fig. 6 Comparisons of community composition for fecal data sets.** The upper panel shows PCA clustering of the results (**A**). Different approaches and tools used for taxonomic annotation (MG - mOTU2, Peptides - Unipept, and Proteins - Prophane) are indicated in the label. Clusters ($k = 3$) were calculated using manhattan distance and are represented by blue, yellow, and green. Features not annotated at species level were considered unclassified and discarded for PCA calculation. Unclassified features accounted for 73.4% and 9.5% of data for peptide and protein subgroup levels. The top 10 variables driving differences between samples are represented by black arrows. The lower panel details taxonomic profiles of each sample as bar plots (**B**). Source data is provided as a Source Data file.

phenotype of the analyzed sample. In order to investigate the influence of post-processing steps on this functional information, we compared functional community profiles on both the SIHU-MIx and the fecal samples (Fig. 7). We observed that the functional similarity between data sets acquired with different workflows on each sample is extremely high, and this regardless of the approach chosen. For the peptide-centric approach, we compared the Gene Ontology (GO) terms (GO domain "biological process") provided by Unipept for each of the identified peptides with MegaGO[64], resulting in MegaGO similarities of 0.96 or higher. Notably, 95% of the identified peptides were associated with at least one GO term. For the protein-centric approach, the protein families (Pfam) annotations provided by

Prophane were compared, resulting in Pearson correlations of 0.98 or higher and Spearman correlations of 0.64 or higher. This continues the trend already observed in Fig. 4: while peptide identifications may differ greatly between samples, the underlying biological meaning reflected by functional annotations are highly similar across different analysis workflows. Moreover, while some more elaborate data measurements yield unique peptides, these peptides do not translate into more functional pathways being identified (Supplementary Fig. 13) and usually correspond to very low abundant proteins, identified with only one peptide (as already shown in Supplementary Fig. 4).

In contrast, a comparison between the different omics domains showed important differences in terms of functional profile.

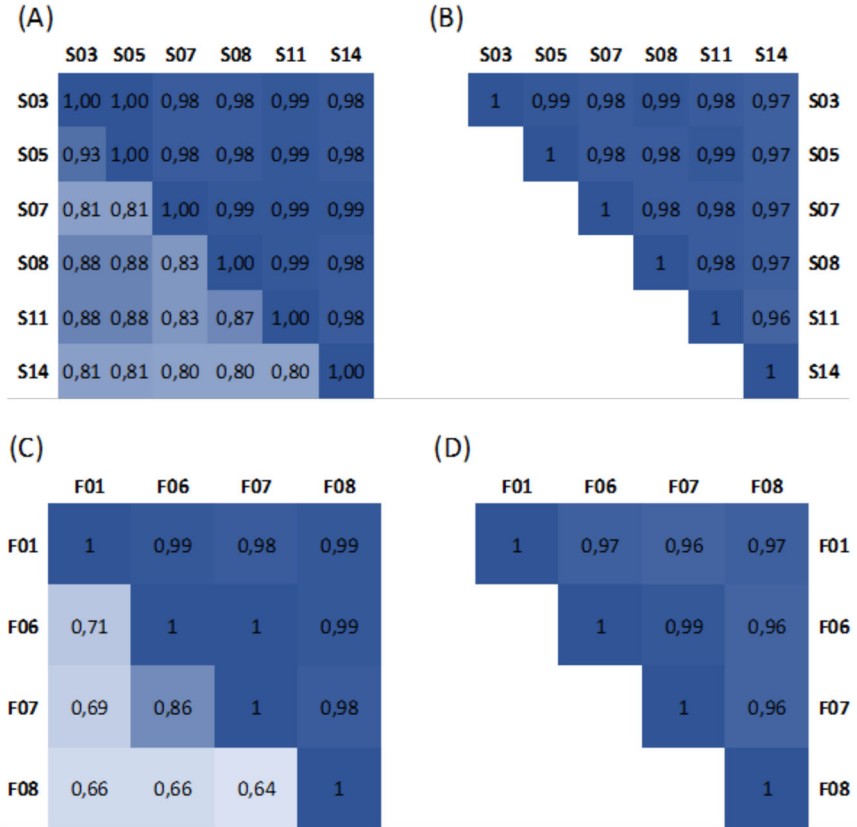

**Fig. 7 Functional similarity between SIHUMIx samples and fecal samples.** The correlation matrices at the left show the Pearson correlation (upper triangle) and Spearman correlation (bottom triangle) for the (**A**) SIHUMIx data sets and (**C**) fecal data sets, calculated using the Pfam annotations returned by the protein-centric Prophane analysis. The correlation matrices at the right show the MegaGO similarity for the GO domain "biological process" for the (**B**) SIHUMIx data sets and (**D**) fecal data sets, calculated based on the GO terms returned by peptide-centric Unipept analyses. Source data is provided as a Source Data file.

Notably, metagenomics and metaproteomics are particularly different from each other, while metatranscriptomics tends to overlap better with metagenomics, highlighting once more the need for integrated meta-omics approaches (Supplementary Figs. 14–16)[32].

## Discussion

In this founding edition of CAMPI, we used both a simplified, laboratory-assembled sample as well as a human fecal sample to compare commonly used experimental methods and computational pipelines in metaproteomics at the peptide, protein subgroup, taxonomic and functional level, informed by and contrasted with metagenomics and metatranscriptomics. Our findings demonstrate some differences in the taxonomic profiles between peptide-centric metaproteomics, protein-centric metaproteomics, and read-based metagenomics, and metatranscriptomics. This fits well with previous findings that assessment of microbial community structure via shotgun metagenomics and metaproteomics differs in the information obtained. While metagenomics has been shown to provide a good representation of per species cell numbers in a community, metaproteomics has been shown to provide a good representation of per species biomass in a community[45]. When looking at different proteomics approaches, differences tend to show up primarily at the finest resolution, such as the sequences of the identified peptide sequences. When considering information from the protein subgroup level up, much of this variation disappears. Different protocols tend to primarily display different levels of analytic

depth, which correlates with more extensive sample fractionation and faster instruments. Moreover, differences between search engines appear somewhat complementary, giving an advantage to integrative, multi-search engine approaches using more sophisticated scoring engines. Interestingly, there appears to be an important contribution to any observed differences from the sequence database used for identification. This is particularly evident in the protein inference step, where peptide-level degeneracy in the database becomes an important factor in the outcome of protein grouping, as already shown and discussed previously[65,66]. Overall, functional profiles of different proteomics workflows were quite similar, which is a reassuring characteristic due to the unique perspective provided by proteomics on the functional level.

Besides the direct conclusions of CAMPI as summarized here, another important outcome of this study is the availability of the acquired data sets. Indeed, these can serve as benchmark data sets for the field when developing novel algorithms and approaches for data processing and interpretation (see "Data availability" section). While it is recommended that researchers use well-annotated matched metagenomes for optimal metaproteomics analysis, not all study designs have metagenomics information available. For such studies, iterative search approaches on publicly available repositories are available[25,67–70], some of which address the issue of controlling the false discovery rate of identifications[68]. Moreover, other platforms such as iMetaLab[30] have been widely used for human and mouse gut metaproteomics analysis. We have not used the iterative search approaches or alternative platforms for this study, although the availability of the data

should encourage users to evaluate the performance of these approaches.

This CAMPI study has highlighted that there is room for future editions of CAMPI studies. Indeed, based on the issues identified in this study, we can already define interesting future research questions: what is the effect of data set complexity, and how do other sample types such as marine sediments affect the results; how is quantification affected by the workflow used, and which quantification approach yields the most robust and accurate results; how are taxonomic resolution, functional profiling, and quantification affected by the dynamic range of the sample composition; and what is the potential of data-independent acquisition (DIA) and targeted approaches in metaproteomics regarding reproducibility and analytical depth?

Obviously, relevant standardized samples will need to be defined for these studies, and should moreover be produced in sufficient amounts to allow their continued use by interested researchers after the publication of these studies. These could take the form of a defined synthetic community with exactly known composition, including cell numbers and sizes, preferably stimulated under different biological conditions. With such a sample, we will be able to validate a variety of quantification methods, but also investigate the effect of quantifying individual proteins in relation to their background. Moreover, it remains a question for now what the effect will be on the taxonomic resolution or functional profile. Label-based approaches could also be extremely valuable for the field as it has been shown that stable isotope labeling as a spike-in reference can strongly improve quantification accuracy[71,72]. On another technical level, we could investigate the opportunities and challenges of the use of DIA on metaproteomics samples. Potentially, there will be new, AI-driven search engines that will enter the field of (meta)proteomics, which also brings new opportunities for the field.

Of course, all these follow-up CAMPI studies will contribute highly useful benchmark samples and data sets to the field as well, thus creating a strong, positive feedback loop with the metaproteomics community. Future CAMPI editions will be launched by the Metaproteomics Initiative (metaproteomics.org), a newly founded community of metaproteomics researchers that aims, among other things, to standardize and accelerate experimental and bioinformatic methodologies in this field. This initiative can combine forces with existing initiatives such as the ABRF iPRG study group, who recently provided a metaproteomics data set to be analyzed by the proteomics informatics community[73]. We believe that such ongoing efforts will continue to advance the field of metaproteomics, and make it more widely applicable. Metaproteomics will thus develop its full potential, and further increase its relevance across the life sciences.

## Methods

**Ethics**. Written informed consent was obtained from the subject enrolled in the study. This study was approved by the ethics committee of the University Magdeburg (reference no. 99/10).

### Sample description

*Simplified human intestinal microbiota sample (SIHUMIx)*. A simplified human intestinal microbiota (SIHUMIx) composed of eight species was constructed to embody a majority of known metabolic activities typically found in the human gut microbiome. The SIHUMIx sample contains the Firmicutes *Anaerostipes caccae* DSMZ 14662, *Clostridium butyricum* DSMZ 10702, *Erysipelatoclostridium ramosum* DSMZ 1402 and *Lactobacillus plantarum* DSMZ 20174, the Actinobacteria *Bifidobacterium longum* NCC 2705, the Bacteroidetes *Bacteroides thetaiotaomicron* DSM 2079, the Lachnospiraceae *Blautia producta* DSMZ 2950, and the Proteobacteria *Escherichia coli* MG1655, covering the most dominant phyla in human feces[74]. SIHUMIx was prepared as previously described, with an additional 24 h of cultivation of one control bioreactor, to produce sufficient biomass to be sent out to each participating laboratory[74]. Participants received $3.5 \times 10^9$ cells/ml of frozen sample ($-20\,°C$) in dry ice.

*Human fecal microbiome sample*. A natural human fecal microbiome sample was procured from a 33-years-old omnivorous, non-smoking woman. The sample was immediately homogenized, treated with RNA-later, aliquoted, frozen, and stored at $-20\,°C$ until aliquots were sent to each participating laboratory.

### Biomolecule extraction and nucleotide sequencing

*DNA/RNA extraction, sequencing, and processing*. DNA was extracted from both SIHUMIx and the fecal samples. RNA could also be extracted from the fecal sample but not SIHUMIx as only the former was treated with RNA-later.

Extracted DNA and RNA were sequenced with Illumina technology, and the obtained sequencing reads subsequently co-assembled into contigs for further bioinformatic processing. Details on the extractions, libraries preparations, and sequencing can be found in Supplementary Note 1. Preprocessing of the sequenced reads was performed as part of the Integrated Meta-omic Pipeline (IMP)[75] and included the trimming and quality filtering of the reads, the filtering of rRNA from the metatranscriptomic data, and the removal of human reads after mapping against the human genome version 38. Preprocessed RNA and DNA reads were co-assembled using MEGAHIT v1.2.4[76] using minimum and maximum k-mer sizes of 25 and 99, respectively, and a k-step of 4. The resulting contigs were binned using MetaBAT 2.12.1[77] and MaxBin 2.2.6[78] with default parameters and minimum contig length of 2500 and 1500 bps, respectively. Bins were refined using DASTool 1.1.2[79] with default parameters and a score threshold of 0.5. Open reading frames (ORFs) were called from all contigs provided to DASTool using Prodigal 2.6.3[80] as part of the DASTool suite.

*Protein extraction and processing*. In total, eight different protein extraction protocols were applied and resulted in 24 different workflows when combined with MS/MS acquisition strategies (Fig. 1). Key characteristics for each workflow can be found in Supplementary Data 1. The most obvious workflow differences were found in protein recovery, cleaning, and fractionation strategies. In a wide comparative approach, the protein extract was processed by either filter-aided sample preparation (FASP)[81] (workflows 1–3, 5, 7–9, 11, 12, 19–23 in Supplementary Data 1), in-gel (workflows 4, 6, 10, 13–18), or in-solution (workflows 21 and 24) digestion. In most workflows, proteins were directly extracted from the raw defrosted material (workflows 1–20, 22, 23). In one lab, however, microbial cells were first enriched at the interface of a reverse iodixanol gradient (workflows 21, 24). In most approaches, cell lysis was based on mechanical cell disruption by bead beating in a variety of chemical buffers (workflows 1–12, 19–23), or in water (workflows 13–18). Apart from bead beating, ultrasonication in a chaotrope-detergent-free buffer was employed to allow for further separation of cytosolic and envelope-enriched microbiome fractions (workflows 21 and 24) and, in another separate workflow, cryogenic grinding was employed for the simultaneous extraction of DNA, RNA, and protein using the Qiagen Allprep kit (workflows 22, 23). Recovery of proteins from the lysis mixture was carried out either by solvent extraction using a variety of solvents, with or without further washes (workflows 4–18, 22, 23), or by filter-aided methods (FASP) (workflows 1–3). All methods included trypsin as the sole proteolytic enzyme for digestion of DTT (or DTE)-reduced and iodoacetamide-alkylated proteins. Digestion was performed either on filters (workflows 1–3, 5, 7–9, 11–12, 19–24), in-gel with or without fractionation (workflows 6, 10, 13–18), or in-solution in the presence of a surfactant (workflows 21 and 24). Of note, the enzyme/substrate ratio varied from 1/50 to 1/10,000, with digestion times from 2 to 16 h. Finally, peptides were recovered from the gel or eluted from filters (FASP) using a salt solution (workflows 1–3, 5–21, 24). In some protocols, peptides were desalted using different commercial devices (workflows 4, 21, and 24).

**LC–MS/MS acquisition**. Each laboratory used its own LC–MS/MS protocol with the largest differences and similarities highlighted in the following and details provided in Supplementary Data 1. For LC, all laboratories separated peptides using reversed-phase chromatography with a linear gradient length ranging from 60 to 460 min. Furthermore, one group performed an additional separation using a multidimensional protein identification technology (MudPIT) combining cation exchange and reversed-phase separation in a single column prepared in-house[82].

Six groups used an Orbitrap mass spectrometer (4× Q Exactive HF, 1× Q Exactive Plus, 1× Fusion Lumos, ThermoFisher Scientific), while two groups employed a timsTOF mass spectrometer (Bruker Daltonik). All participants used data-dependent acquisition (DDA) with exclusion duration times ranging from 10 to 60 s. All MS proteomics data and X!Tandem results have been deposited to the ProteomeXchange Consortium (http://proteomecentral.proteomexchange.org) via the PRIDE partner repository[83].

### Bioinformatics

*Generation of protein sequence databases*. Two types of databases were used for each sample; a catalog (reference) database and a database that was generated from metagenomic and metatranscriptomic (when available) data sequenced from a matching sample (meta-omic database). The catalog database for SIHUMIx consisted of the combined reference proteomes of the strains extracted from UniProt in July 2019[84] except for *Blautia producta*, for which the whole genus *Blautia* was taken (SIHUMIx_REF). The IGC 9.9 database[85] (available at http://meta.genomics.

cn/meta/dataTools) was used as the catalog database for the fecal sample (GUT_REF). Additionally, a meta-omic database from the assembled contigs was produced for both samples using the open reading frame generated with Prodigal (SIHUMIx_MO and GUT_MO).

The SIHUMIx database (SIHUMIx_REF) is composed of reference proteomes, containing 29,557 proteins (13.2 MB). In comparison, the metagenomic assembly for SIHUMIx (SIHUMIx_MO) produced 2719 contigs, with an average contig length of 7.5 Kbp and the longest contigs being 468 Kbp, yielding 19,319 predicted ORFs (6.1 MB).

For the fecal sample, the IGC reference catalog (GUT_REF) contains 9,879,896 protein sequences (2.6 GB). The co-assembly of DNA and RNA for the fecal sample (GUT_MO) produced 247,518 contigs with an average length of 1.6 Kbp and the longest contigs being 600 Kbp. The database GUT_MO yielded protein sequences from 441,558 predicted ORFs (114.4 MB). All databases were concatenated with a cRAP database of contaminants (https://thegpm.org/cRAP; downloaded in July 2019) and the GUT databases were additionally concatenated with the human UniProtKB Reference Proteome (downloaded in September 2019).

The four databases were in silico digested into tryptic peptides with an in-house developed script, with two missed cleavages allowed, to compare their theoretical search spaces. Additionally, all peptides identified with each database in the explorative analysis, which was carried out using all data sets, were retrieved and compared.

For metaproteomic data analysis, the number of spectra, PSMs, and identification rates (calculated by dividing the number of identified spectra by the total number of acquired MS/MS spectra) were extracted for all data sets searched against the selected databases (SIHUMIx_REF and GUT_MO) and compared. Finally, a representative subset of data sets, based on the different methods, was selected for further analysis (S03, S05, S07, S08, S11, S14 for SIHUMIx and F01, F06, F07, and F08 for the fecal sample).

*Data analysis using four different bioinformatic pipelines.* All submitted MS/MS raw files were first analyzed with a single commonly used database search method to assess both the quality of the extraction and the MS/MS acquisition, as well as the effect of the search database composition (reference proteomes vs. multi-omics). For this, X!Tandem[51] (Alanine, 2017.02.01) was used as a search engine with the following parameters: specific trypsin digest with a maximum of two missed cleavages; mass tolerances of 10.0 ppm for MS1 and 0.02 Da for MS2; fixed modification: Carbamidomethylation of C (+57.021464 Da); variable modification: Oxidation of M (+15.994915 Da); fixed modification during refinement procedure: Carbamidomethylation of C (+57.021464 Da). Peptides were filtered by length (between 6 and 50 amino acids), and charge state (+2, +3, and +4), and a maximum valid expectation value (*e*-value) of 0.1[86].

The following database search engines were used for the pipeline comparison: (i) MaxQuant[87] (including the search engine Andromeda) (ii) Galaxy-P workflows[88,89] consisting of SearchGUI[90,91] (using OMSSA[92], X!Tandem[51], MS-GF+[59], and Comet[93]) and PeptideShaker[94] to merge the results, (iii) MetaProteomeAnalyzer[28] (server version 3.4, using X!Tandem and OMSSA), and (iv) ProteomeDiscoverer 2.2 (using SequestHT, from ThermoFisher). The identification settings for all search engines were the same as for the explorative analysis mentioned above. Refinement searches were allowed if implemented in the search engine (e.g., refinement search of X!Tandem), and the same for the inclusion of post-processing tools (e.g., Percolator within ProteomeDiscoverer).

*Protein inference.* To allow protein group comparison, groups were created using the combined peptide evidence of all compared samples. Two different protein grouping methods were tested: MPA[28] and PAPPSO[61], and analyses were made on protein groups and subgroups (Supplementary Note 3).

Assigning peptides to their correct protein can be a difficult task, notably due to the protein inference issue[3], i.e., the same peptide can be found in different homologous proteins. This is particularly challenging in metaproteomics where the diversity and number of homologous proteins are much higher compared to single-species proteomics. To overcome this issue, most bioinformatic pipelines tend to automatically group homologous protein sequences into protein groups. However, each tool handles protein inference and protein groups in its own way, which prevents a straightforward output comparison at the protein group level. In order to allow robust comparison between approaches, the PSM output files of the four bioinformatic pipelines were combined. The peptides were then assigned to protein sequences in the FASTA file and the data was prepared for subsequent protein grouping. Two approaches of protein grouping were used and evaluated in this study: PAPPSO grouping[61], which excludes proteins based on the rule of maximum parsimony, and grouping from MPA[28], which does not exclude proteins. All data processing was done using a custom Java program except for PAPPSO grouping for which data was exported and imported using the appropriate XML format.

For both methods, protein groups were created using the loose rule "share at least one peptide" (groups) and the strict rule "share a common set of peptides" (subgroups), resulting in a total of four protein grouping analyses: (1) PAPPSO groups, (2) MPA groups, (3) PAPPSO subgroups, and (4) MPA subgroups. Finally, the resulting protein groups and subgroups were exported for further analysis (Supplementary Note 3). These algorithms are also implemented in Pout2Prot[95] for independent use.

*Taxonomic and functional annotation.* Annotations were performed at both the peptide, protein, and the sequencing read level. Unipept was used for the peptide-centric approach[24,27,96]. For the taxonomic annotation of the SIHUMIx data sets, we used an advanced Unipept analysis that calculates the SIHUMIx-specific lowest common ancestor (LCA) (i.e., it calculates the LCA specific for its search database instead of the complete UniProtKB). Here, Unipept searched for the occurrence of each peptide in all species present in NCBI. For each peptide separately, we removed those species that cannot be present in the SIHUMIx sample (i.e., non-SIHUMIx species and contaminating species in the cRAP database), after which we calculated the SIHUMIx-specific LCA. This advanced taxonomic analysis using Unipept is possible since the composition of the sample is known, and resulted in a more accurate taxonomic annotation of the peptides. For more information and examples of the advanced Unipept analysis (Supplementary Note 4). For the taxonomic annotation of the fecal data sets with Unipept, the desktop[96] and CLI[23,97] versions were used. In both analyses for SIHUMIx and the fecal data sets, isoleucine (I) and leucine (L) were equated. The assigned taxonomies for each of the peptides can be found in Supplementary Data 8 and 9.

For the functional analysis at the peptide level, we used the Unipept command line option to extract the GO terms for each identified peptide per data set (below 1% FDR). The functional similarity of these sets of GO terms was calculated with MegaGO[64].

Prophane was used for the protein-centric approach[98,99]. For both the functional and taxonomic annotations, a generic output format created by the in-house developed protein grouping script and the protein database for a given analysis were used. Within Prophane, the taxonomic annotation was performed with DIAMOND blastp against the latest NCBI non-redundant (nr) database (2019-09-30)[100], while two functional annotation tasks where performed against the eggNOG (database version 4.5.1)[101] and Pfam-A (db version 32) databases[102] using eggNOG-mapper[103,104] and hmmscan[105], respectively. Using eggNOG-mapper, the e-value threshold was set to 0.0005 while we applied a gathering threshold supported by Pfams (cut_ga parameter) when searching using hmmscan. The result with the protein group identifiers from the previous analysis summary can be found in Supplementary Data 10–12, and the assigned taxonomies for each of the proteins can be found in Supplementary Data 13 and 14.

Metagenomic and metatranscriptomic reads were both taxonomically annotated with the mOTUs profiler v 2.0[106] with default parameters at the species and family levels for SIHUMIx and the feces sample, respectively.

Quantification was based on read counts for metagenomic and metatranscriptomics data, and on spectral counts for peptides and protein subgroups. If two subgroups contained the same peptide, spectra would be counted twice, distorting the abundance of these particular subgroups inside a measurement, but preserving a consistent count for comparison with other samples. Comparisons were performed with normalized values as described in detail below.

*Comparison between omics domains—taxonomic resolution.* Taxonomic annotations from the Prophane protein group outputs were used for metaproteomics. This method uses only identified proteins and assesses annotations based on the LCA approach thus generating results for each protein at the best possible taxonomic resolution.

The mOTU2 profiler used for the metagenomic taxonomic annotation takes advantage of marker genes for taxonomic annotation and thus annotates everything at the OTU level. Since this approach does not allow comparison at each taxonomic level, Kraken2[107] was used to compare taxonomic resolution across omics domains. Kraken2 was run on the sequencing reads with the maxikraken2_1903 database and a confidence threshold set to 0.7.

*Comparison between omics domains - functional comparison.* Each sequence database (SIHUMIx_REF, SIHUMIx_MO, and GUT_MO) was annotated with the Mantis[108] tool for consensus-driven protein annotation. For metaproteomics, abundance from Prophane outputs and annotation from Mantis were used to generate functional profiles. For metagenomics and metatranscriptomics, sequencing reads were mapped against the assembly contigs using bowtie2[109] and ORFs abundance was calculated using featureCounts[110] KEGG[111] annotations were retrieved from Mantis and used to compare functional profiles across omes.

**Statistical analyses**. Differences and overlap between search engines at the peptide level and between approaches at the peptide level using presence/absence data were visualized with UpSet plots with the UpSetR package[112]. For the peptides, sequences were extracted (without modifications and with leucine (L) and isoleucine (I) treated equally and replaced by J) from each result file and a table, indicating whether a peptide was found or not, was prepared (Supplementary Note 4 and Supplementary Data 15 and 16). Similar tables and UpSet plots were generated to visualize differences and overlap between sample preparations for the peptides, the protein subgroups, and the top 50% protein subgroups. The top 50% were first selected based on abundance data. The spectral counts were summed for each subgroup across all selected samples and only the top 50% was kept for UpSet plot comparison. Results from the taxonomic annotations for all approaches (peptides, proteins, metagenomic and metatranscriptomic reads) were compared and visualized using the PCA comparison feature of the R prcomp package. For the comparison, abundance values (number of reads and spectral counts) were used

and normalized into percentage. The taxonomic annotations were harmonized across methods, unclassified values were filtered out and annotations with abundance lower than 0.05% after filtering were grouped into "other".

All correlation plots were calculated using both Pearson and Spearman correlations with a *p*-value < 0.001. The correlations were calculated and plotted using the corrplot R packages.

Hierarchical clusterings were calculated with the R function hclust using the Manhattan distance and the Ward method.

**Reporting summary**. Further information on research design is available in the Nature Research Reporting Summary linked to this article.

## Data availability
The metaproteomic data sets generated and analyzed in the current study are available via the PRIDE partner repository with the data set identifier PXD023217. Assemblies and raw metagenomic and metatranscriptomic reads are available through the European Nucleotide Archive under the study accession number PRJEB42466. Source data are provided with this paper.

## Code availability
All scripts and intermediary files are made available on github.com/metaproteomics/CAMPI[113].

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

## Acknowledgements

This work has benefited from collaborations facilitated by the Metaproteomics Initiative (https://metaproteomics.org/) whose goals are to promote, improve and standardize metaproteomics. Part of the LC–MS/MS measurements were made in the Molecular Education, Technology, and Research Innovation Center (METRIC) at North Carolina State University, the ProGénoMIX platform at CEA-Marcoule supported by the IBISA network. Parts of the bioinformatics analysis were carried out using the high-performance computing facilities of the University of Luxembourg (https://hpc.uni.lu). This work was supported by the Research Foundation - Flanders (FWO) [grant no. 1S90918N (S.B.) to T.V.D.B.; 12I5217N to B.M.; G042518N to L.M.]; by a FEBS Summer Fellowship [to T.V.D.B.]; by the European Union's Horizon 2020 Program (H2020-INFRAIA-2018-1) [823839 to L.M.]; by the FEMS [R.T.G. to S.S.S.]; by the Norwegian Centennial Chair program [to T.J.G., P.D.J., and M.A.]; the Novo Nordisk Foundation grant NNF20OC0061313 to M.A.; the USDA National Institute of Food and Agriculture Hatch project [1014212 to M.K.]; the U.S. National Science Foundation [OIA 1934844 and IOS 2003107 to M.K.]; the Foundation for Food and Agriculture Research [Grant ID: 593607 to M.K.]; the Agence Nationale de la Recherche [ANR-17-CE18-0023-01 to G.M., O.P., J.A.]; Deutsche Forschungsgemeinschaft (DFG) [RE3474/5-1 and RE3474/2-2 to S.F., T.M., B.Y.R.]. Research by T.J.G., P.D.J, E.L was funded by National Cancer Institute-Informatics Technology for Cancer Research (NCI-ITCR) grant 1U24CA199347 and National Science Foundation (U.S.) grant 1458524 to T.J.G.; and the National Institutes of Health R01-DK70977 to RLH. The European Galaxy server that was used for some calculations is in part funded by Collaborative Research Centre 992 Medical Epigenetics (DFG grant SFB 992/1 2012) and the German Federal Ministry of Education and Research (BMBF grants 031 A538A/A538C RBC, 031L0101B/031L0101C de.NBI-epi, 031L0106 de.STAIR, 031L0103 MetaProtServ (de.NBI)). This work was supported by the Luxembourg National Research Fund (FNR) under grants PRIDE/11823097 and CORE-INTER/13684739 to B.J.K., P.M. and P.W., and the European Research Council (ERC-CoG 863664) to P.W.

## Author contributions

T.V.D.B., B.J.K., K.S. and S.S.S. contributed equally as the main authors of the manuscript; doing the data curation, data analysis and visualization, and writing and editing the manuscript. N.J. and D.B. started and supervised the initial laboratory intercomparison in the context of the 3rd International Metaproteomics Symposium. L.M. and T.M. started and jointly supervised the process of turning the intercomparison study into a coherent manuscript and supervised the writing. T.V.D.B., B.J.K., K.S., S.S.S., L.M., T.M., N.J. and D.B. contributed to the data selection, concept, and structure of the manuscript. B.J.K., S.S.S., J.-P.T, R.H., M.W., R.H., M.K., M.J., G.M., O.P., J.A., R.L.H., S.L.P., R.J.G., C.J., C.H., M.V.B., P.E.A. and T.L. did laboratory work for one of the eight independent laboratories, which included the sample provision, distribution and pre-paration, protein extraction and proteolytic digestion, mass spectrometry and nucleotide sequencing. T.V.D.B., K.S., P.J., R.H., M.W., M.Ø.A., L.H.H., H.S., S.F., M.J., O.P., S.L.P., R.J.G., A.B., K.T., E.L., A.S., P.T.Q., P.V., P.M., B.M., P.E.A. and O.L. did bioinformatics analyses that were used in the manuscript, which included the generation of protein databases from nucleotide sequences, mass spectrometry data processing and initial protein database search, further data analysis, and meta-analysis and visualization. J.A., R.L.H., P.J., C.L., M.K., O.P., R.H., P.T.Q., B.M., A.T., S.F. and P.M. reviewed and edited the manuscript and participated in discussions on the course of individual analysis. J.A., R.L.H., P.J., C.J., M.K., M.Ø.A., U.R., B.Y.R., S.U., M.V.B., P.W. and T.J.G. had a supervisory role and reviewed the manuscript. The four main authors T.V.D.B., B.J.K., K.S. and S.S.S. are listed at the beginning, while the two supervising authors L.M. and T.M. are listed at the end. All other co-authors are listed alphabetically.

## Competing interests

The authors declare no competing interests.

## Additional information

¹VIB - UGent Center for Medical Biotechnology, VIB, Ghent, Belgium. ²Department of Biomolecular Medicine, Faculty of Medicine and Health Sciences, Ghent University, Ghent, Belgium. ³Luxembourg Centre for Systems Biomedicine, University of Luxembourg, Esch-sur-Alzette, Luxembourg. ⁴Bioprocess Engineering, Otto-von-Guericke University Magdeburg, Magdeburg, Germany. ⁵Department of Molecular Systems Biology, Helmholtz-Centre for Environmental Research - UFZ GmbH, Leipzig, Germany. ⁶Biosciences Division, Oak Ridge National Laboratory, Oak Ridge, TN, USA. ⁷Département Médicaments et Technologies pour la Santé (DMTS), Université Paris Saclay, CEA, INRAE, SPI, 30200 Bagnols-sur-Cèze, France. ⁸Faculty of Chemistry, Biotechnology and Food Science, Norwegian University of Life Sciences (NMBU), Ås, Norway. ⁹INRAE, AgroParisTech, Micalis Institute, Université Paris-Saclay, 78350 Jouy-en-Josas, France. ¹⁰Microbiology, Department of Applied Biosciences and Process Technology, Anhalt University of Applied Sciences, Köthen, Germany. ¹¹Bioprocess Engineering, Max Planck Institute for Dynamics of Complex Technical Systems, Magdeburg, Germany. ¹²Bioinformatics Unit (MF1), Department for Methods Development and Research Infrastructure, Robert Koch Institute, Berlin, Germany. ¹³Department of Biochemistry Molecular Biology and Biophysics, University of Minnesota, Minneapolis, MN, USA. ¹⁴Department of Plant & Microbial Biology, North Carolina State University, Raleigh, USA. ¹⁵Université Paris-Saclay, INRAE, CNRS, AgroParisTech, GQE – Le Moulon, 91190 Gif-sur-Yvette, France. ¹⁶Department of Applied Mathematics, Computer Science and Statistics, Ghent University, Ghent, Belgium. ¹⁷Data Analytics and Computational Statistics, Hasso-Plattner-Institute, Faculty of Digital Engineering, University of Potsdam, Potsdam, Germany. ¹⁸Faculty of Technology, Bielefeld University, Bielefeld, Germany. ¹⁹Department of Biomedical Sciences, University of Sassari, Sassari, Italy. ²⁰Integrated Biobank of Luxembourg, Luxembourg Institute of Health, 1, rue Louis Rech, L-3555 Dudelange, Luxembourg. ²¹Department of Life Sciences and Medicine, Faculty of Science, Technology and Medicine, University of Luxembourg, 6 avenue du Swing, L-4367 Belvaux, Luxembourg. ²²Section eScience (S.3), Federal Institute for Materials Research and Testing, Berlin, Germany. ²³These authors contributed equally: Tim Van Den Bossche, Benoit J. Kunath, Kay Schallert, Stephanie S. Schäpe. ²⁴These authors jointly supervised this work: Lennart Martens, Thilo Muth. ✉email: lennart.martens@ugent.be

