## [Peer Review File · Nature Communications]

REVIEWER COMMENTS

Reviewer #1 (Remarks to the Author):

The manuscript of Tim Van Den Bossche and colleagues is a very interesting multi-lab test finalized at the development of standardized guidelines for metaproteomics investigations.

The manuscript is well written, original and of scientific sound; nevertheless, the reader of this manuscript would benefit from the Authors expertise while facing less common samples such as the food associated microbiota, marine sediments and so on.

Isolation of the microbial fraction out of complex matrices is of great importance for the entire metaproteomics workflow; thus, the author's vision on this very first step would be useful for the novel readers approaching this fascinating field.

Similarly, the wide instrumental choice is likely to influence the whole metaproteomics survey. A discussion on this aspect would be useful along with the author's suggestion on the best choice between the label-based and the label-free approaches.

Finally, the bioinformatic data analysis described in the present manuscript deals with the different approaches and search engines for protein ID inference. Heterogeneous results might also be recorded during the following steps of functional classification of the identified proteins depending on the data repository of choice and the availability of sequences and sequence annotation completeness.

Can authors deepen these aspects and, if applicable, provide their suggestion on the most suitable data repositories to be queried while retrieving the functional featuring of the microbial communities?

Also, it should be noted that although of great importance for the overall metaproteomics community, the manuscript does not details the composition and/or activities of the investigated samples; thus, its publication could be more suitable for methodological journals since dealing with methods optimization.

Reviewer #2 (Remarks to the Author):

In this work, the authors carried a comprehensive assessment of metaproteomics workflows between different laboratories. This work also demonstrated the robustness of metaproteomics. The results from this collaboration are valuable for the researchers who want to build/improve their own metaproteomics workflows.

However, the manuscripts often presents oversimplification of the results which led to simplified conclusions that are not taking based on exploring the depth of the data. For example, metaproteomics is probably robust across the labs for higher abundance proteins, but less robust for lower abundance proteins. Our suggestions include:

The bioinformatic analysis is limited to one metaproteomic software. Other approaches have been previously reported and would be simple to incorporate. In particular, I suggest adding the multistage searching strategy which was shown to improvement the identification rates in metaproteomics.

More information is needed for the comparison between multi-omics results. Currently not even a Venn diagram showing the identified species from metaproteomics and metagenomics is provided;

The functional annotation section needs to be reorganized and highlighted. As the authors said, functional annotation is a unique advantage of metaproteomics approaches. Comparison with metagenomics/metatranscriptomics results is required.

Page 10, line 252-254. "Here, these public reference databases (SIHUMIx_REF and GUT_REF) contained 1.6 and 16 times, respectively, more unique in silico digested peptides than the corresponding multi-omic databases".

Since the increase in database size from GUT_MO to GUT_REF is much bigger than from SIHUMIx_MO to SIHUMIx_REF, why the decrease of identification rates in some samples are similar (S9, S10, S11 suffer about 6% losses, and F5, F6, F7 loss similar numbers)? I'm just curious , because the size of SIHUMIx_REF is only 29,557 proteins. Although the SIHUMIx_MO database could be more appropriate and refined, considering the small size of this database, the losses of 6% identification rates are strange. Could you use SearchGUI/PeptideShaker here to see what will happen?

Page 10, paragraph 2. The discussion about the database usage is decent and reasonable. However, this part is about the wet-lab workflows, which requires more discussion on the influence of various experimental techniques, and their positive/negative effects on the results. Some recommendations should be provided here. I understand that for different experimental aims/conditions, there is no universal "best approach" for all of the situations. However, based on the results, I thought the samples S09, S10, S11, F05, F06, F07 from the same lab gave better results. Please add some description about their workflow and the corresponding advantages. Otherwise, samples S13, S14, F08, F09 didn't perform very well here, the possible reasons/optimization should also be discussed.

Page 11, line 294-298. "16% of the peptides were uniquely identified by a single bioinformatic pipeline for the S11 data set (3138, 2670, 891, and 841 peptides for SearchGUI/PeptideShaker, MaxQuant, Proteome Discoverer, and MPA, respectively), while this was 27% for the F07 data set (6024, 1264, 819, and 332 peptides for the same pipelines, respectively)".

For the F07 dataset, do you mean 6024 for SearchGUI/PeptideShaker, 1264 for MaxQuant, 819 for Proteome Discoverer and 332 for MPA? The numbers are different from the numbers in Figure 3B.

Page 11, line 298-300. "The number of search engines varies between pipelines, with one for MaxQuant (Andromeda) and ProteomeDiscoverer (SequestHT), two for MPA (X!Tandem, OMSSA), and four for SearchGUI (X!Tandem, OMSSA, MS-GF+, and Comet)".

In my opinion, only MPA is a specially designed tool for metaproteomics data analysis, and the other three search engines are general methods in proteomics data analysis. The results obtained from searching metaproteomics datasets indicate that the MPA is not the best. In theory, MPA using multiple search engines should perform better than MaxQuant and PD. Generally, this part is about the bioinformatic workflows used in metaproteomics analysis. The authors should use other bioinformatic pipelines especially designed for metaproteomics. I recommended the adoption of multistage searching strategy for the comparison which can handle large microbial protein database.

Page 13, Figure 4. The results from 6 SIHUMIx samples were selected for the comparison here and in Figure 4. The common part is limited to about 5%, is this because some samples have lower numbers of identifications? For example, if S03 and S07 are excluded, will the common part increase significantly? I think the conclusion on the robustness of metaproteomic need to be revised as the authors did not use all of the results from the participating labs to reach their conclusions.

Page 15, line 394-397. "For the SIHUMIx samples, the two protein grouping methods PAPPSO and MPA provided very similar numbers of both protein groups and subgroups".

Please show the specific numbers here.

Page 16, line 418. "we quantitatively compared the identified taxa for each selected sample".

Is this quantitative information from peptide count, spectra count or peptide intensities? Please make it clear. If it's based on the peptide count or spectra count, could you use peptide intensity as a comparison?

Page 16, line 441. "Additionally, a false negative was observed for *Clostridium butyricum*..."

Why is this one a false negative? I think this is a good result, even if it is not observed in the read-based method. I checked the data and found 27 unique peptides specific to *Clostridium butyricum*, so I think it's a confident identification.

Page 17, line 442-443. "while two false positives were observed for Clostridiales bacterium and *Bacteroides dorei*".

Please provide the tables as supplementary files showing the "peptides to taxa" and "proteins to taxa" information. The result tables from Unipept and Prophan directly should be fine. Do you add some filters, e.g., peptide count threshold or protein abundance threshold, to remove the species with lower confidence?

Page 18, line 451-452. "Unclassified features accounted for 24.2% and 69.9% of data for peptide and protein subgroup levels".

So the comparison between metaproteomics and metagenomics methods for the taxonomy analysis is less informative. The differences of compositions shown here are mainly because of the loss of quantitative information in the species level from the metaproteomics methods. I think it would be better to do the comparison at the genus level to include more information from metaproteomics methods.

Page 18, paragraph 2. The comparison of the number of identified taxa and the overlaps between the three methods, metagenomics, metatranscriptomics and metaproteomics should be shown here. It will be better to show the data in different taxonomic levels. I'd like to know from which level, the metaproteomics workflow will have obviously worse taxonomic resolution than metagenomics method.

Page 21, paragraph 1. The functional annotation part should be rewritten because too many details are missed here. I totally agreed with the authors' opinion, that the "major strength of metaproteomics is the ability to provide functional information that reflects the phenotype of the analyzed sample". This major strength needs to be better displayed. First of all, a figure containing the information about the COG/NOG categories, COG/NOG composition or KEGG pathways need to be provided. Second, the comparison between multi-omics functional annotation results is needed to be discussed here. Third, more detailed comparison between different workflows is also necessary. Currently only the similarity between different workflows were illustrated which highlight the robustness of metaproteomics. However, this is likely a narrow view of the results based on the high abundance proteins. Clearly, some of the results from the different labs were not as sensitive. What are the differences between these results in terms of composition, function and pathways?

Reviewer #1 (Remarks to the Author)

- 1. The manuscript of Tim Van Den Bossche and colleagues is a very interesting multi-lab test finalized at the development of standardized guidelines for metaproteomics investigations.**

Answer: We thank the Reviewer for their time and effort reviewing our manuscript, and for the overall positive assessment of our work.

- 2. The manuscript is well written, original and of scientific sound; nevertheless, the reader of this manuscript would benefit from the Authors expertise while facing less common samples such as the food associated microbiota, marine sediments and so on.**

Answer: This is a very interesting comment, which we will definitely take into account for the next CAMPI editions. On June 15, we presented this study at the online International Metaproteome Symposium, and this suggestion also came up there. Therefore, we have added this suggestion in the Discussion of our revised manuscript: "Indeed, based on the issues identified in this first study, we can already define interesting future research questions: what is the effect of data set complexity, and how do other sample types such as marine sediments affect the results; (...)".

- 3. Isolation of the microbial fraction out of complex matrices is of great importance for the entire metaproteomics workflow; thus, the author's vision on this very first step would be useful for the novel readers approaching this fascinating field.**

Answer: Indeed, in metaproteomics, the initial step of extracting protein biomass from a sample is critical to accurately capture the full protein complement of the microbial community. Environmental samples, such as feces or soil, are complex mixtures that can contain microbial cells, host cells, plant-derived fibrous materials, and other abiotic components. Therefore, composition and abundance of these components must be considered when choosing an appropriate method for cellular lysis and protein extraction. Fortunately, the most commonly used methods nowadays are relatively robust, and generally provide a reasonably representative extraction of proteins found in these complex mixtures. Furthermore, additional

enrichment and/or depletion strategies can be employed to provide better detection of the microbial representation in the presence of complicating environmental components/interferences. However, the current study was not set up to achieve this level of granularity in the analysis of the effects of the (many) different workflow steps. At the same time, it is a very good suggestion for a subsequent CAMPI study, which could focus on these important first steps.

We therefore added a part of this (longer) answer into the introduction of our revised manuscript and added relevant literature for novel readers.

4. Similarly, the wide instrumental choice is likely to influence the whole metaproteomics survey. A discussion on this aspect would be useful along with the author's suggestion on the best choice between the label-based and the label-free approaches.

Answer: Instrument choice is indeed a big contributing factor for differences between laboratory workflows as indicated by the numbers in Figure 2. However, we would propose that the choice of MS instrument specifically is typically not as important, as it is a large one-time investment a laboratory cannot improve immediately and is widely discussed in the broader proteomics community. However, as we cannot exclude the influence of the mass spectrometer on the final results, we added the following sentence in the introduction of our revised manuscript: "Besides, apart from different sample processing protocols, different mass spectrometers might also lead to a variation in results.". In contrast, other laboratory parameters such as the question of label-free and label-based approaches brought up by the reviewer are highly relevant for any reader that attempts to make choices for upcoming projects. Unfortunately, our analysis only includes label-free analyses, also because label-based approaches are uncommon in metaproteomics so far. Yet, again, as above, future CAMPI studies could include label-based approaches. Therefore, we included the following sentence in the Discussion of our revised manuscript: "Label-based approaches could also be extremely valuable for the field as it has been shown that stable isotope labelling as a spike-in reference can strongly improve quantification accuracy(Zhang et al. 2016; von Bergen et al. 2013)."

5. Finally, the bioinformatic data analysis described in the present manuscript deals with the different approaches and search engines for protein ID inference. Heterogeneous results might also be recorded during the following steps of functional classification of the identified proteins depending on the data repository of choice and the availability of sequences and sequence annotation completeness. Can authors deepen these aspects and, if applicable, provide their suggestion on the most suitable data repositories to be queried while retrieving the functional featuring of the microbial communities?

Answer: We tried to separate the different stages of bioinformatic analysis in this manuscript as well as possible. In particular, we used the tools Unipept and Prophaner to achieve taxonomic and functional annotation independently of the search engines used. The underlying sequence databases and annotation ontologies will of course have an impact on the taxonomic annotation. In section 4.4 we pointed out that taxonomic annotation is achieved based on the UniProtKB/Swiss-Prot database when using Unipept and on the NCBI nr database when using Prophaner. However, keep in mind that Unipept also uses NCBI taxonomies. We performed an additional analysis on functional annotation by adding annotation from metagenomics. For this analysis, we used Mantis (Queirós et al. 2021) that provides consensus protein annotation using multiple data repositories such as Pfam, KEGG, GO, and EC. We found Mantis to provide the most comprehensive annotations for the protein sequences, as it can take advantage of multiple data repositories. We present the results of Mantis annotations in the supplementary information of the revised manuscript, but also for the omic comparison that was requested by Reviewer 2 (comment 13).

6. Also, it should be noted that although of great importance for the overall metaproteomics community, the manuscript does not details the composition and/or activities of the investigated samples; thus, its publication could be more suitable for methodological journals since dealing with methods optimization.

Answer: While no biological insight was attempted in this study, this, of course, was also not the objective, and the study was also not designed for that purpose. However, while our manuscript may focus on method analysis and optimization, it

also introduces the concept of standardized, controlled data sets for community wide-use in, for instance, benchmarking. Moreover, our study shows the fundamental robustness of present-day metaproteomics approaches, and thus should satisfy the requirements for Nature Communications, which, according to the guide to authors, aims to publish "...high-quality papers from all areas of science that represent important advances within specific scientific disciplines". Again, we do believe that this target has been met here. Indeed, this manuscript describes the first, founding edition of CAMPI which is of great importance for the scientific community interested in microbiota analysis.

Reviewer #2 (Remarks to the Author):

- 1. In this work, the authors carried a comprehensive assessment of metaproteomics workflows between different laboratories. This work also demonstrated the robustness of metaproteomics. The results from this collaboration are valuable for the researchers who want to build/improve their own metaproteomics workflows. However, the manuscripts often presents oversimplification of the results which led to simplified conclusions that are not taking based on exploring the depth of the data. For example, metaproteomics is probably robust across the labs for higher abundance proteins, but less robust for lower abundance proteins. Our suggestions include:**

The bioinformatic analysis is limited to one metaproteomic software. Other approaches have been previously reported and would be simple to incorporate. In particular, I suggest adding the multistage searching strategy which was shown to improvement the identification rates in metaproteomics.

More information is needed for the comparison between multi-omics results. Currently not even a Venn diagram showing the identified species from metaproteomics and metagenomics is provided;

The functional annotation section needs to be reorganized and highlighted. As the authors said, functional annotation is a unique advantage of metaproteomics approaches. Comparison with metagenomics/metatranscriptomics results is required.

Answer: We thank the reviewer for the summary of suggestions. These points are repeated in detail in the comments below, so we have addressed them for clarity over there.

- 2. Page 10, line 252-254. "Here, these public reference databases (SIHUMIx_REF and GUT_REF) contained 1.6 and 16 times, respectively, more unique in silico digested peptides than the corresponding multi-omic databases".**

Since the increase in database size from GUT_MO to GUT_REF is much bigger than from SIHUMIx_MO to SIHUMIx_REF, why the decrease of identification rates in some samples are similar (S9, S10, S11 suffer about 6% losses, and F5, F6, F7 loss similar numbers)? I'm just curious , because the size of SIHUMIx_REF is only 29,557 proteins. Although the SIHUMIx_MO database could be more appropriate and refined, considering the small size of this database, the losses of 6% identification rates are strange. Could you use SearchGUI/PeptideShaker here to see what will happen?

Answer: We have checked the average (and standard deviation) of the difference between the identification rates (in %) between the MO and REF databases. The difference for the SIHUMIx samples is 4.12% +/- 1.63% and for the fecal samples 5.03% +/- 1.41%. Placed in that perspective, the 6% losses seem therefore still acceptable. Interestingly, we see that this particular loss of 6% for both databases is found for the samples coming from the same laboratory (S09, S10, S11, F5, F6, and F7 are all from the same group). As suggested by the reviewer, the files were imported in PeptideShaker to check the QC plots, but all of them were correct and cannot explain the 6% loss. Even though we do not have the final answer to this puzzling question, the fact that the QC check did not reveal anything and that all samples were provided by the same group suggest that this difference is due to the extraction method rather than the database. We hypothesized that the extraction used for S09, S10, S11, F5, F6, and F7 might have a particular difference that would lead to such loss but no obvious differences could be determined.

3. Page 10, paragraph 2. The discussion about the database usage is decent and reasonable. However, this part is about the wet-lab workflows, which requires more discussion on the influence of various experimental techniques, and their positive/negative effects on the results. Some recommendations should be provided here. I understand that for different experimental aims/conditions, there is no universal “best approach” for all of the situations. However, based on the results, I thought the samples S09, S10, S11, F05, F06, F07 from the same lab gave better results. Please add some description about their workflow and the corresponding advantages. Otherwise, samples S13, S14, F08,

F09 didn't perform very well here, the possible reasons/optimization should also be discussed.

Answer: We agree with the Reviewer that this is important as well. These wet-lab recommendations were pointed out at the second paragraph on page 9. We have added the performance increasing methods between brackets in that paragraph: "The differences in acquired spectra show a clear relation to the method used, as similar methods or replicates show highly similar numbers of acquired spectra. As expected, more complex methods with longer gradient lengths (S03 and S04: 260 min, S05 and S06: 460 min, S08: 240 min, F01: 210 min, F02: 160 min), fractionation (S11, F07: 4 fractions), and additional separation methods such as MudPIT (F01: 4 fractions) or ion mobility (PASEF) (S13, F09) led to up to eight times more identified spectra, but at the cost of increased time and resources spent (see Supplementary Table 1 for a detailed description, and Supplementary Table 2 for an overview of the samples)." The rest of the original paragraph already makes additional further comparisons.

4. Page 11, line 294-298. "16% of the peptides were uniquely identified by a single bioinformatic pipeline for the S11 data set (3138, 2670, 891, and 841 peptides for SearchGUI/PeptideShaker, MaxQuant, Proteome Discoverer, and MPA, respectively), while this was 27% for the F07 data set (6024, 1264, 819, and 332 peptides for the same pipelines, respectively)".

For the F07 dataset, do you mean 6024 for SearchGUI/PeptideShaker, 1264 for MaxQuant, 819 for Proteome Discoverer and 332 for MPA? The numbers are different from the numbers in Figure 3B.

Answer: Thank you for pointing out this mistake, the figures are actually correct and the order changes between SIHUMIx and the fecal sample, which is now correctly addressed in the text: "Interestingly, 16% of the peptides were uniquely identified by a single bioinformatic pipeline for the S11 data set (3138, 2670, 891, and 841 peptides for SearchGUI/PeptideShaker, MaxQuant, Proteome Discoverer, and MPA, respectively), while this was 27% for the F07 data set (6024, 1264, 819, and 332 peptides for the SearchGUI/PeptideShaker, Proteome Discoverer, MPA and MaxQuant pipeline, respectively)."

5. Page 11, line 298-300. “The number of search engines varies between pipelines, with one for MaxQuant (Andromeda) and ProteomeDiscoverer (SequestHT), two for MPA (X!Tandem, OMSSA), and four for SearchGUI (X!Tandem, OMSSA, MS-GF+, and Comet)”.

In my opinion, only MPA is a specially designed tool for metaproteomics data analysis, and the other three search engines are general methods in proteomics data analysis. The results obtained from searching metaproteomics datasets indicate that the MPA is not the best. In theory, MPA using multiple search engines should perform better than MaxQuant and PD. Generally, this part is about the bioinformatic workflows used in metaproteomics analysis. The authors should use other bioinformatic pipelines especially designed for metaproteomics. I recommended the adoption of multistage searching strategy for the comparison which can handle large microbial protein database.

Answer: We thank the reviewer for this comment. MPA is indeed the only pipeline *intended* for metaproteomics, however, MPA also uses standard search engines originally intended for classical, single-species proteomics (specifically, MPA uses OMSSA and X!Tandem). The main focus of MPA is to provide user-friendly downstream analysis and annotation features for taxonomies and functions, not to provide a specialised search engine. However, we did not even use this particular functionality of MPA, instead opting for two more commonly used tools (Unipept and Prophan) to assign such taxonomy and function information to the identified peptides and proteins from all four upstream bioinformatic pipelines for a fair comparison.

We also tried iMetaLab, but we couldn't get this tool to provide any results. After repeatedly trying to contact the developers, but without any success, we decided not to include iMetaLab in the manuscript. Moreover, there were also no submissions with iMetaLab, nor participants who used iMetaLab in the study.

Regarding the multistage search strategy or two-step database search: while indeed commonly applied, it actually sacrifices FDR control. The two-step approach as originally described in <https://onlinelibrary.wiley.com/doi/full/10.1002/pmic.201200352>, generates a reduced

database using results of the search against the initial database. But it has been shown by Muth et al (<https://onlinelibrary.wiley.com/doi/full/10.1002/pmic.201400560>) that this forfeits FDR control. Therefore, knowing that it might return a larger amount of false positive identifications than expected based on FDR control, we did not include the two-step search strategy, and moreover fear that it might create too many error propagation issues in the downstream taxonomic and functional analysis.

6. Page 13, Figure 4. The results from 6 SIHUMix samples were selected for the comparison here and in Figure 4. The common part is limited to about 5%, is this because some samples have lower numbers of identifications? For example, if S03 and S07 are excluded, will the common part increase significantly? I think the conclusion on the robustness of metaproteomic need to be revised as the authors did not use all of the results from the participating labs to reach their conclusions.

Answer: This is an excellent point and something we thought about during our analysis. It first needs to be noted that six and four samples were selected for SIHUMix and the fecal samples respectively, because of their diversity in workflow. This is the first analysis of this kind in metaproteomics and one of the challenges was indeed the question of how to deal with such a diverse set of samples. By selecting only some samples we tried to reduce the amount of samples, while also retaining the most interesting differences (different mass spectrometer, fractionation, etc.). We also believe that the robustness of metaproteomics is better highlighted at the functional and taxonomic level, even with this diverse set of samples.

Still we would like to address the specific comment. Figure 4 from our original manuscript (see below) only shows the intersections of size one (only found in a single sample) and of maximum size (found in all samples).

Figure 4. UpSet plot comparison of sets of identified peptides (A and B), protein subgroups (C and D), and 50% most abundant protein subgroups based on spectral counts (E and F). The figure is based on the identifications obtained using SearchGUI/PeptideShaker. The intersection size displays the number of features shared in an intersection. An intersection corresponds to features shared across multiple samples. This figure only displays features unique to a sample (red dot), and shared across all samples (blue bar overlapping all points).

We included in our original manuscript more comprehensive UpSet plots in Supplementary Figure 1 and 2, that were omitted from the main text in the original manuscript for the sake of readability. In these plots, the first twenty intersections are ordered by their size showing actual counts. For the Reviewer's convenience, we copied the two figures below. There is indeed some effect from the number of

samples included, for instance in Supplementary Figure 1B excluding samples S07 and S14 would increase the intersection of the other four samples from 2312 to 3666 (519+440+395), certainly a substantial increase. In general, however, the trend we described is even more obvious: on the peptide level (Figure 4A), identifications unique to a single sample are dominant, on the protein subgroup level (Figure 4B), this effect is decreased with the intersection of all samples rising to the top, while on the level of Top-50% protein subgroups identifications unique to a single sample virtually disappear. Furthermore, in Supplementary Figure 2 of our original manuscript we plotted the effect of continuously removing the lowest abundance protein subgroups. We also included the actual spectral count for each intersection (dashed line), showing that in terms of actual abundance, the difference between samples is much smaller, i.e. ~77% of spectra belong to protein subgroups that are found in every sample. This is similar for both SIHUMIx and the fecal sample, despite using a different number of samples for comparison (6 and 4 respectively).

Taken together, these data led us to the conclusion that the differences between samples are caused by the random identification of low abundance proteins: proteins that typically have only one or two peptide spectrum matches identifying them. Because the samples contain so many low abundance proteins at the limit of detection, each measurement will lead to the identification of a different set of these low abundance proteins. Therefore, adding more samples to a comparison will increase the disagreement on the peptide level, but our analysis suggests that all samples will agree on the higher abundance proteins that also constitute a large portion of peptide spectrum matches.

Suppl. Figure 1. Complete UpSet plot comparison for the SIHUMix dataset. The different panels show the comparison at the levels of A) peptides, B) protein subgroups and C) top 50% of the protein subgroups between sample preparations. Unlike Figure 4, the UpSet plot shows the 20 most abundant intersections, regardless of the intersection composition. For Peptides, the unique intersections (size=1) are the most prominent, while they entirely disappear for the top 50% protein subgroups. S03 and S05 are almost identical laboratory workflow, differing in LC gradient length, and the intersection between S03 and S05 is large for the peptides.

Suppl. Figure 2. Complete UpSet plot comparison for the FECES dataset. The different panels show the comparison at the levels of A) peptides, B) protein subgroups and C) top 50% of the protein subgroups between sample preparations. Unlike Figure 4, the UpSet plot shows the 20 most abundant intersections, regardless of the intersection composition. For Peptides, the unique intersections (size=1) are the most prominent, while they entirely disappear for the top 50% protein subgroups. F06 and F07 are almost identical laboratory workflow, one of them utilizing fractionation, and the intersection between F06 and F07 is large for the peptides.

7. Page 15, line 394-397. “For the SIHUMIx samples, the two protein grouping methods PAPPSO and MPA provided very similar numbers of both protein groups and subgroups”.

Please show the specific numbers here.

Answer: We added the numbers as requested: “For the SIHUMIx samples, the two protein grouping methods PAPPSO and MPA provided very similar numbers of both protein groups (8802 and 8769) and subgroups (10132 and 10134), while substantial differences were found for the fecal samples (protein groups: 10063 and 9712; subgroups: 17576 and 21973, PAPPSO and MPA respectively) (Supplementary Table 4)”. The complete list of numbers can, as indicated in text, be found in Supplementary Table 4.

8. Page 16, line 418. “we quantitatively compared the identified taxa for each selected sample”.

Is this quantitative information from peptide count, spectra count or peptide intensities? Please make it clear. If it’s based on the peptide count or spectra count, could you use peptide intensity as a comparison?

Answer: We used spectral counting. We agree that this was not clearly addressed in the original manuscript (mentioned only in the description of Figure 4). We therefore changed the sentence accordingly in our revised manuscript:

“To determine if differences between workflows have an effect on the overall biological conclusions, we quantitatively compared the identified taxa for each selected sample from both data sets using spectral counts, and this at the peptide, the protein subgroup, and the sequencing read level.”

Moreover, we acknowledge that spectral count is not seen as a state-of-the-art quantification method in single-species proteomics, but for the first CAMPI edition we focussed on the identification of peptides and protein groups, rather than its detailed quantification. However, although it’s not seen as a state-of-the-art quantification method, recently it was shown by Kleiner et al that the differences between spectral counts and peptide intensities were negligible, and that peptide intensities could even underestimate the abundances of some species. Therefore, we decided that spectral counts would be the safer option for now [<https://www.nature.com/articles/s41467-017-01544-x>]. Moreover, because the

measurements took place in different laboratories and different mass spectrometers, the comparison of the MS1 would not be straightforward at all. This does, however, bring up new ideas for the follow-up CAMPI edition, as already addressed in the Discussion and future outlook section of the original manuscript:

“Indeed, based on the issues identified in this first study, we can already define interesting future research questions: (...); how is quantification affected by the workflow used, and which quantification approach yields the most robust and accurate results”.

9. Page 16, line 441. “Additionally, a false negative was observed for *Clostridium butyricum*...”

Why is this one a false negative? I think this is a good result, even if it is not observed in the read-based method. I checked the data and found 27 unique peptides specific to *Clostridium butyricum*, so I think it’s a confident identification.

Answer: We agree with the Reviewer that this might be confusing. We meant that *Clostridium butyricum* was not identified by the read-based method, and that this species therefore presents a false negative identification in the read-based method, not the proteomic methods. Therefore, to make it clearer to the readers, we changed the sentence to the following in the revised manuscript:

“Additionally, *Clostridium butyricum* was not found by the read-based method, while *Clostridiales* bacterium and *Bacteroides dorei* were falsely found by the protein-centric method as these are not present in the SIHUMIx sample.”

10. Page 17, line 442-443. “while two false positives were observed for *Clostridiales* bacterium and *Bacteroides dorei*”.

Please provide the tables as supplementary files showing the “peptides to taxa” and “proteins to taxa” information. The result tables from Unipept and Prophaner directly should be fine. Do you add some filters, e.g., peptide count threshold or protein abundance threshold, to remove the species with lower confidence?

Answer: We included these files as supplementary material of our revised manuscript. For the peptides to taxa files, see Supplementary Files 3 and 4). For the

proteins to taxa files, see Supplementary Files 5 and 6. The references to these Supplementary Files were placed in Methods section 4.4 of the revised manuscript. We did not use additional filters to remove species with lower confidence.

11. Page 18, line 451-452. “Unclassified features accounted for 24.2% and 69.9% of data for peptide and protein subgroup levels”.

So the comparison between metaproteomics and metagenomics methods for the taxonomy analysis is less informative. The differences of compositions shown here are mainly because of the loss of quantitative information in the species level from the metaproteomics methods. I think it would be better to do the comparison at the genus level to include more information from metaproteomics methods.

Answer: We believe that in a lab-assembled mixture, it should be possible to go as deep as the species level. However, we already included the analysis on genus level in Supplementary Table 5 in the original manuscript, but now updated this Supplementary Table 5 in the revised manuscript with a more extensive analysis. Here, we observe that at the genus level, we are not able to identify 8.6% of the peptides: 1.3% find no classification back in Unipept, while the rest is annotated at a higher lca. In addition, we added the taxonomic distribution and principal component analysis (PCA) plot as Supplementary Figure 9. For the Reviewers' convenience, we added the Figures also below. In the main text, we however kept the species level as the sample is fully described and thus provides an unbiased view of taxonomic resolution efficiency. We added the following sentence at the bottom of that paragraph: “For completeness, the comparisons of community composition for SIHUMix at the genus level were added in Supplementary Figure 9.”

Supplementary Figure 9. Comparisons of community composition for SIHUMix on genus level. The upper panel shows PCA clustering of the results. Different approaches and tools used for taxonomic annotation (mOTU2, Unipept and Prophan) are indicated in the label. Clusters (k=3) were calculated using Manhattan distance and are represented by blue, yellow, and green. Features not annotated at species level were considered unclassified and discarded for PCA calculation. Unclassified features accounted for 24.2% and 69.9% of data for peptide and protein

subgroup levels. Variables driving differences between samples are represented by black arrows. The lower panel details taxonomic profiles of each sample as bar plots.

12. Page 18, paragraph 2. The comparison of the number of identified taxa and the overlaps between the three methods, metagenomics, metatranscriptomics and metaproteomics should be shown here. It will be better to show the data in different taxonomic levels. I'd like to know from which level, the metaproteomics workflow will have obviously worse taxonomic resolution than the metagenomics method.

Answer: This comment is difficult to address due to the different tools used for each omics domain. For metagenomics, we used a gene-marker based tool (mOTU2) and thus 100% of the sequences used for taxonomic profiling have a resolution at the mOTU level. In order to have information at each taxonomic level, we could use another tool such as Kraken2. However, using Kraken2 generates another type of issue because it uses the entire set of metagenomic reads. Many reads can be of low information (e.g low complexity sequences, less conserved regions, sequences from horizontal gene transfer events,...) and thus lead to a lot of unidentified reads. In contrast, in metaproteomics, only the proteins that were identified/expressed are used for taxonomic profiling, reducing already drastically the set size and focusing on more informative/better described sequences (coding sequences that are expressed). Because of all those differences, when comparing taxonomic resolution between metagenomic and metaproteomic, we obtain two contrasting figures. For the first, using the gene-marker based method, 100% of the reads considered are annotated at the best resolution possible. For the second (using the all-read set method) many reads contain no taxonomic information. So even though we know from previous work that metagenomics would have a better taxonomic resolution than metaproteomics, it is not what we would observe when comparing metaproteomics with the output from Kraken2.

In the following figure (also added as Supplementary Figure 11 A/B of the revised manuscript), we compared those outputs for both SIHUMIx (a) and the fecal samples (b).

Supplementary Figure 11. A/B: Taxonomic resolution across omics domains for SIHUMIx (a) and the Fecal (b) samples. The stacked bar plots represent the percentage of metagenomic (MG) and metatranscriptomic (MT) reads or identified proteins (MPA and PAPPSO) that can be annotated at the Phylum (P), Class (C), Order (O), Family (F), Genus (G) and Species (S) levels. For metaproteomics, the percentage of annotation starts to decrease at the Genus level and becomes very low at the Species level. For the complex fecal sample (b), PAPPSO grouping allows better taxonomic resolution as already described in the Supplementary Note 1.3. For metagenomics and metatranscriptomics, the lower resolution is also higher at the species level but already appears at the Phylum level. As detailed previously, this is

mostly due to the method used to allow such comparison. Indeed we annotated every single read with Kraken2, while the metaproteomics approach only focused on identified proteins. This important difference in size of the data and the fact that metaproteomics analyses a subset of it can justify the differences observed here.

13. Page 21, paragraph 1. The functional annotation part should be rewritten because too many details are missed here. I totally agreed with the authors' opinion, that the "major strength of metaproteomics is the ability to provide functional information that reflects the phenotype of the analyzed sample". This major strength needs to be better displayed. First of all, a figure containing the information about the COG/NOG categories, COG/NOG composition or KEGG pathways need to be provided. Second, the comparison between multi-omics functional annotation results is needed to be discussed here. Third, more detailed comparison between different workflows is also necessary. Currently only the similarity between different workflows were illustrated which highlight the robustness of metaproteomics. However, this is likely a narrow view of the results based on the high abundance proteins. Clearly, some of the results from the different labs were not as sensitive. What are the differences between these results in terms of composition, function and pathways?

Answer: We agree with the Reviewer that indeed some points were missing in the original manuscript. For the revised manuscript, we re-annotated the protein sequences from each database using Mantis. Mantis is a newly developed tool for consensus annotation using different annotation databases like eggNOG, KEGG, and EC. We fetched KEGG annotations (KOs) for all sequences and compared their identifications and abundances at the metagenomic and metaproteomic level (and metatranscriptomic when available).

To answer the first question, we retrieved the identified KEGG for each omic domain and mapped them on the "Microbial metabolism in diverse environments" KEGG map in order to display the overlaps and differences between different omics domains.

KEGG ID retrieved for SIHUMIx using the metagenomic database was compared to the combined metaproteomic results in the attached Supplementary Figure 12 of the revised manuscript.

Supplementary Figure 12. Mapping of the KEGG ID onto the Microbial metabolism in diverse environments KEGG map for SIHUMIx. Metagenomics, metaproteomics and the overlap are displayed in green, blue and purple, respectively.

As expected, we can observe that the expressed functions found in the metaproteomics data correspond to a subset of those identified in the metagenomics data representing the complete potential of the community and mostly overlap it. Few functions are found only by metaproteomics, but this is due to the different database used for the SIHUMIx metaproteomic analysis. Indeed, the reference database was used for the search, showing that few functions were missed during the generation of the metagenomic database.

Similar comparison was performed for the fecal dataset between metagenomics, metatranscriptomics and metaproteomics, shown in the Supplementary Figure 13 of the revised manuscript.

Supplementary Figure 13. Mapping of the KEGG ID onto the Microbial metabolism in diverse environments KEGG map for the fecal dataset.

Metagenomics, metatranscriptomics and metaproteomics are displayed in green, yellow and blue, respectively. Overlap between metagenomics and metatranscriptomics is coloured in brown, overlap between metagenomic and metaproteomics is coloured in purple and overlap between all omics domains is coloured in black.

Similarly to SIHUMIx, the pathways identified in the metaproteomics data are mostly a subset of the metagenomic pathways. Interestingly, very few functions were uniquely found via metatranscriptomics (yellow) but a good part of the functions found by metagenomics are covered by metatranscriptomics (brown) but not by metaproteomics. This indicates that the functional potential identified by metatranscriptomics is closer to the one identified by metagenomics than the one from metaproteomics and that the latter is lacking depth of analysis.

To go further into the comparison and add the quantitative aspect, we retrieved the read mapping counts for each ORF at the metagenomic/metatranscriptomic level and spectral counting for each protein subgroup at the metaproteomic level. We retrieved the KEGG annotations and compared the omic domains using PCA

(Supplementary Figures 14 of the revised manuscript). Similarly to the PCA plots for the taxonomic profiles, we can observe differences between omics domains, with metagenomics being relatively distant to the other omics domains and metaproteomics and metatranscriptomics being relatively close to each other. As expected, the differences at the functional level are much stronger than at the taxonomic level. Indeed, when a taxon switches metabolism, no differences will be observed in the taxonomic profile of the community, but the functional profile will be affected, thus leading to bigger differences at the functional level.

Supplementary Figure 14A. Comparisons of community's functional composition for SIHUMix. The PCA analysis shows a strong separation between metagenomics (green) and the different metaproteomic samples (blue).

Supplementary Figure 14B. Comparisons of community's functional composition for the FECAL sample. The PCA analysis shows a strong separation between metagenomics (green) and the different metaproteomic samples (blue). Metatranscriptomics (yellow) is in between the two other omics domains, but very similar to metagenomics.

Finally, for the latest part of the question, we compared the samples with the highest (S11 and F01) and lowest number (S07 and F06) of identified protein subgroups for both SIHUMix and the FECAL datasets.

As already shown in Figure 4 of the original manuscript, each method brings its own uniqueness, with some approaches being deeper than others and producing more MS/MS spectra. For SIHUMix, S07 has only 1.7% of unique protein subgroups against 8,1% for S11 (Figure 4C). Similar observations can be made in Supplementary Figure 15 of the revised manuscript where almost no functions are found uniquely in S07 in contrast to S11.

The fecal samples show similar results (Supplementary Figure 15B of the revised manuscript), with the small difference that the least sensitive approach (F06) has more uniquely found functions associated as already discussed in Figure 4. Indeed, the more complex the sample is, the more marked the differences are between each method.

It is interesting to note that no new pathways were identified by the most sensitive methods in comparison to the least sensitive ones. Instead, similar pathways are identified but with a bigger coverage of functions. The only exception is for SIHUMIx, where S11 identifies a partial “purine metabolism” that is missed by S07.

Finally, it should be noted that even though there are differences in pathway completeness in terms of identification, those are minimal when the quantification is taken into account. As shown in Supplementary Figure 3 of the original manuscript, the uniquely identified protein subgroups for all datasets are usually identified with only one peptide and account for less than 4% of the total spectra.

Supplementary Figure 15A. Mapping of the KEGG ID onto the ‘Microbial metabolism in diverse environments’ KEGG map for SIHUMIx. The most (S11) and the least (S07) sensitive approaches are displayed in green and pink, respectively. The overlap is shown in purple.

Supplementary Figure 15B. Mapping of the KEGG ID onto the 'Microbial metabolism in diverse environments' KEGG map for the Fecal dataset. The most (F01) and the least (F06) sensitive approaches are displayed in green and pink, respectively. The overlap is shown in purple.

REVIEWERS' COMMENTS

Reviewer #1 (Remarks to the Author):

The authors have neatly improved the manuscript, responding extensively to the proposed questions; however, there remains some perplexity regarding the target audience of the paper.

I agree with the authors that microbiota is of extreme interest for the scientific community, but it is also true that they present methods and not biological 'story'.

However, I postpone the final decision to the Editor, being understood that the paper is very good, even if strongly methodological.

Reviewer #2 (Remarks to the Author):

The authors have addressed most of my comments. I think that the overall conclusions of the paper are sound. It is nevertheless, a paper from a group of scientists that use a common bioinformatic approach to identify peptides.

The authors indicated that they attempted to use iMetaLab, but were not successful in implementing this approach. I am surprised as it is used by many other labs. Nevertheless, other tools are available. The issues of higher false discovery has been reported using the two-step approach. However, more recent strategies have addressed this issue. I think that the lack of alternative search approach is a limitation of this paper and it needs to be pointed out in the discussion section.

Reviewer #1

The authors have neatly improved the manuscript, responding extensively to the proposed questions; however, there remains some perplexity regarding the target audience of the paper.

I agree with the authors that microbiota is of extreme interest for the scientific community, but it is also true that they present methods and not biological 'story'.

However, I postpone the final decision to the Editor, being understood that the paper is very good, even if strongly methodological.

Answer: We thank the Reviewer for their efforts in reviewing our manuscript.

Reviewer #2

The authors have addressed most of my comments. I think that the overall conclusions of the paper are sound. It is nevertheless, a paper from a group of scientists that use a common bioinformatic approach to identify peptides.

The authors indicated that they attempted to use iMetaLab, but were not successful in implementing this approach. I am surprised as it is used by many other labs. Nevertheless, other tools are available. The issues of higher false discovery has been reported using the two-step approach. However, more recent strategies have addressed this issue. I think that the lack of alternative search approach is a limitation of this paper and it needs to be pointed out in the discussion section.

Answer: We thank the Reviewer for their comments. At the time of working on this manuscript, iMetaLab was primarily used for human/mouse gut microbiome research. We have added a paragraph in the discussion of the second revision of the manuscript to address this. We have also mentioned the other approaches available for large database searches and mentioned this as a limitation in this manuscript.

“While it is recommended that researchers use well-annotated matched metagenomes for optimal metaproteomics analysis, not all study designs have metagenomics information available. For such studies, iterative search approaches on publicly available repositories are available (PMID: 23412978, PMID: 26084232, <https://www.biorxiv.org/content/10.1101/605550v6>, PMID: 32396365, PMID: 30525664), some of which address the issue of controlling false discovery rate of identifications (PMID: 32396365, PMID: 26084232). Moreover, other platforms such as iMetaLab (PMID: 29912378) have been widely used for human and mouse gut metaproteomics analysis. We have not used the iterative search approaches or alternative platforms for this study, although the availability of the data should encourage users to evaluate the performance of these approaches.”